# GRADCFG: GRADIENT INVERSION OF CLASSIFIER-FREE GUIDANCE DIFFUSION MODELS

## ABSTRACT

Gradient inversion attacks, as a means of privacy theft, have been extensively studied and applied in classifier models, yet research on gradient inversion for diffusion models, particularly classifier-free guidance (CFG) diffusion models, remains relatively underdeveloped. CFG models such as Stable Diffusion present significant challenges for such attacks due to their complex training mechanisms, including the high-dimensional search space caused by multimodal variables, the non-uniqueness of the noise $\epsilon$ solution space, and the difficulty in optimizing discrete time steps $t$. To address these challenges, this paper proposes a novel joint inversion framework featuring two core algorithmic innovations: the **GradCFG** algorithm, which integrates a four-variable co-optimization mechanism for simultaneous reconstruction of image latent variables $\mathbf{x}_0$, text embeddings $C_0$, noise $\epsilon$, and reparameterized continuous time steps $t$, alongside a periodic restart strategy for $\epsilon$ to enhance solution stability and generalization; and the **Inv-Sam** algorithm, a model-difference-based generation optimization method that leverages the generative capability disparities between pre-fine-tuning and post-fine-tuning models to restore high-resolution details through a reverse-forward diffusion editing process. Systematic experiments in CFG model fine-tuning scenarios demonstrate that the proposed method effectively achieves high-quality image-text joint reconstruction for various textual conditions ranging from concise descriptions to complex semantic combinations.

## 1 INTRODUCTION

Diffusion models (Ho et al., 2020) have achieved remarkable breakthroughs in the field of image generation by transforming random noise into high-fidelity images through a progressive denoising process. Classifier-Free Guidance (CFG) (Ho & Salimans, 2022) further introduces a text-conditioning mechanism, enabling semantically controllable image synthesis. Representative CFG models such as Stable Diffusion (SD) (Rombach et al., 2022) have been widely adopted in industry. With the growing demand for personalized generation, users often fine-tune pre-trained models on private data (Gal et al., 2022; Kumari et al., 2023; Hu et al., 2021; Bahmani et al., 2022). To protect privacy, users typically share only training gradients with the server instead of the original images (Sun et al., 2021). However, this process still faces severe privacy leakage risks: malicious attackers can reconstruct private training samples from the gradients via Gradient Inversion Attacks. Such attacks have been extensively demonstrated in classification models (Hatamizadeh et al., 2022; Zhu et al., 2019; Wei et al., 2020), yet the unique training mechanism of diffusion models provides inherent defense capabilities. In particular, for CFG models, the text-guided mechanism further increases the difficulty of gradient inversion.

The challenge of performing gradient inversion on CFG-based diffusion models stems from their multimodal and stochastic training pipeline. During training, the client samples a Gaussian noise vector $\epsilon$ and a time step $t$ independently, and computes gradients from both the image latent $\mathbf{x}_0$ and the text embedding $C_0$. Consequently, the attacker must recover a coupled four-variable tuple $(\mathbf{x}_0, C_0, \epsilon, t)$ instead of a single image, substantially enlarging the solution space. Our analysis further shows that the sampled noise $\epsilon$ does not have a unique solution—multiple $\epsilon$ values can satisfy gradient alignment when accompanied by appropriately adjusted $(\mathbf{x}_0, C_0, t)$. This non-uniqueness requires the remaining variables to flexibly adapt to feasible noise solutions rather than converge to a single fixed point. At the same time, the discrete nature of the time step $t$ prevents direct gradient-based

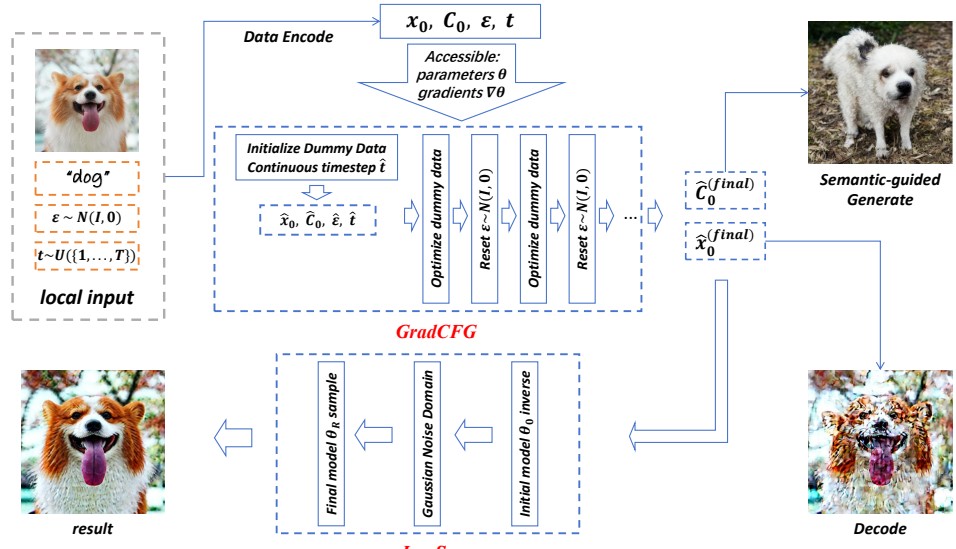

Figure 1: The GradCFG method reconstructs the original data through gradient matching, employing a time-step $\hat{t}$ continuous strategy and a periodic $\hat{\epsilon}$ reset mechanism to mitigate variable discreteness and solution non-uniqueness. Furthermore, the Inv-Sam method refines the initially reconstructed image by leveraging the generative capability gap between the initial and final state models, thereby recovering richer image details.

updates, making the joint optimization even more challenging. Moreover, given that the CFG model is already a well-trained generative model, we are particularly interested in exploring how its inherent generative capability can be incorporated into the reconstruction process to further enhance the quality and detail of the recovered results.

To address these challenges, this paper proposes an innovative solution consisting of two interconnected algorithmic contributions, as shown in Figure 1. First, we propose the **GradCFG** algorithm, which constructs a four-variable joint optimization framework that reparameterizes the discrete time step $t$ as a continuous variable $\hat{t}$, enabling simultaneous optimization of $\hat{x}_0$, $\hat{C}_0$, $\hat{\epsilon}$, and $\hat{t}$. Through periodic restart and re-optimization of $\hat{\epsilon}$, the GradCFG algorithm allows $\hat{x}_0$, $\hat{C}_0$, and $\hat{t}$ to achieve gradient alignment across different $\hat{\epsilon}$ values. Second, we develop the **Inv-Sam** algorithm, a model-disparity optimization strategy that leverages the difference in generative capabilities between the pre-trained model at initial and final fine-tuning stages. This approach runs a reverse–forward diffusion process guided by the pre-trained and fine-tuned models respectively, strategically injecting model-disparity information into the reconstruction pipeline to significantly enhance textual alignment and semantic adaptation in recovered images.

Experiments are conducted under personalized fine-tuning scenarios for CFG models, employing a DREAMBOOTH-like (Ruiz et al., 2023) fine-tuning paradigm to systematically evaluate the method's effectiveness. Our comprehensive evaluation framework encompasses both general textual prompts and specific textual prompts during fine-tuning, enabling a thorough analysis of reconstruction performance across different semantic granularities. We simultaneously assess the recovery quality of both image data and textual embeddings, providing a holistic evaluation of multimodal privacy leakage. Additionally, we conduct ablation studies to systematically investigate the specific impact of the periodic restart mechanism on reconstruction fidelity, while also designing controlled experiments to validate the non-uniqueness of solutions for the noise variable $\epsilon$. The proposed approach is rigorously validated on multiple fine-tuning datasets containing diverse semantic categories, demonstrating robust performance across varying image contents and textual descriptions.

The core contributions of this paper are as follows:

- For the first time, we empirically demonstrate a joint image-text privacy leakage attack in text-guided diffusion models (CFG), opening a new attack surface.

- We propose the first joint optimization framework that simultaneously reconstructs images, text, noise, and time steps, overcoming the challenge of variable coupling under complex training mechanisms.
- We innovatively leverage the generative capability disparity of diffusion models during training to design a reconstruction optimization method, significantly enhancing the detail restoration and visual quality of high-resolution images.

## 2 RELATED WORK

**Classifier-Free Guidance (CFG) Models and Privacy**. Diffusion model variants integrating CFG have become mainstream architectures in text-to-image generation (Rombach et al., 2022; Dhariwal & Nichol, 2021; Saharia et al., 2022; Ramesh et al., 2022). Current research on CFG model privacy primarily focuses on Membership Inference Attacks (Shokri et al., 2017), Model Inversion Attacks (Zhou et al., 2024), and Training Data Extraction (Carlini et al., 2023), while privacy leakage risks from **gradient inversion attacks** during training remain systematically underexplored.

**Gradient Inversion**. Early pioneering work (e.g., DLG proposed by Zhu et al. (2019)) first demonstrated the feasibility of reconstructing training data from gradients, building upon earlier recognition of gradients as a primary leakage channel in collaborative learning (Podschwadt et al., 2022). The attack was primarily effective for small batches (e.g., size=1) and low-resolution images. Subsequent research significantly enhanced its practicality and scope: Geiping et al. (2020) improved reconstruction quality on complex datasets (e.g., ImageNet) by introducing a cosine similarity loss and critical regularizations like Total Variation (Zhu & Blaschko, 2021); Zhao et al. (2020) developed an analytical method to deduce labels from gradients exactly; Yin et al. (2021) proposed GradInversion, which leveraged Batch Normalization statistics and group consistency to tackle larger batches and higher-resolution images. Recently, methodologies have diversified: Generation-based attacks (GEN-GIA) (Wei et al., 2020; Jeon et al., 2021) employ pre-trained generative models (e.g., GANs, diffusion models) as strong priors to produce high-fidelity reconstructions, but their reliance on external data and sensitivity to architectures limit generality. Analytics-based attacks (ANA-GIA) (Gao et al., 2021; He et al., 2019) derive data through maliciously altering model parameters or analyzing model outputs, which is efficient but operates under a strong threat model. In addition to direct attacks, Tian et al. (2025) explored reconstructing data by analyzing the weight differences between pre-training and post-training states of models. Notably, Huang et al. (2025a) investigated gradient inversion in diffusion models, but their approach fails to address the fundamental challenges of sampling noise $\epsilon$ multiplicity and time step $t$ discontinuity, while also being constrained by dependency on pre-trained generators. Currently, research on gradient inversion attacks targeting diffusion models, particularly their widely adopted CFG mechanism, remains notably scarce (Yu et al., 2024).

## 3 METHODOLOGY

This paper focuses on the fine-tuning scenario of CFG models, employing a training paradigm **similar to DREAMBOOTH** (Ruiz et al., 2023). The objective is to reconstruct private training data—including images and their corresponding text embeddings—from gradient information.

### 3.1 CFG MODEL FINE-TUNING FRAMEWORK

In this distributed training framework, each client (user) maintains four private data elements during training round $r$: Raw image $X$, Text prompt $P$, Sampled time step $t \sim \mathcal{U}(\{1, \ldots, T\})$, Gaussian noise $\epsilon \sim \mathcal{N}(\mathbf{0}, \mathbf{I})$. None of these elements are directly shared with the server. The fine-tuning process is shown in Algorithm. 1. The training objective minimizes the following denoising loss function:

$$\mathcal{L}(\theta_r) = \mathbb{E}_{t, \boldsymbol{x}_0, \epsilon} \left[ \left\| \epsilon - \epsilon_{\theta_r} \left( \sqrt{\bar{\alpha}_t} \boldsymbol{x}_0 + \sqrt{1 - \bar{\alpha}_t} \epsilon, t, C_0 \right) \right\|^2 \right], \tag{1}$$

where the frozen pretrained image encoder $\text{Vae}(\cdot)$ encodes the raw image $X$ into latent representation $\boldsymbol{x}_0 = \text{Vae}(X)$, and the text encoder $\text{Encoder}(\cdot)$ maps the text prompt $P$ to conditional embedding $C_0 = \text{Encoder}(P)$; $\theta_r$ denotes the learnable parameters of the diffusion model at round $r$, $\bar{\alpha}_t$ is the hyperparameter controlling the noise schedule in the forward diffusion process, and $\epsilon_{\theta_r}(\cdot)$ represents the noise prediction network taking $(\cdot)$ as input.

---

**Algorithm 1** CFG Model Training Process

---

**Input:** Training rounds $R$, user dataset $\mathcal{X}$, text prompt $P$, diffusion steps $T$, learning rate $\eta$
Pretrained model $\theta_0$, pretrained text encoder Encoder, pretrained VAE $\mathrm{Vae}(\cdot)$
**for** *training round* $r = 1, 2, \ldots, R$ **do**

> **User execution:**
> Encode image: $\boldsymbol{x}_0 \leftarrow \mathrm{Vae}(X)$
> Encode text: $C_0 \leftarrow \mathrm{Encoder}(P)$
> Sample time step: $t \sim \mathcal{U}(\{1, \ldots, T\})$
> Sample noise: $\boldsymbol{\epsilon} \sim \mathcal{N}(\mathbf{0}, \boldsymbol{I})$
> Compute loss function: $\mathcal{L}(\theta_r) = \mathbb{E}_{t,\boldsymbol{x}_0,\boldsymbol{\epsilon}} \left[ \left\| \boldsymbol{\epsilon} - \boldsymbol{\epsilon}_{\theta_r} \left( \sqrt{\bar{\alpha}_t} \boldsymbol{x}_0 + \sqrt{1 - \bar{\alpha}_t} \boldsymbol{\epsilon}, t, C_0 \right) \right\|^2 \right]$
> Compute gradient: $g^{(r)} = \nabla_{\theta_r} \mathcal{L}(\theta_r)$
> Send $g^{(r)}$ to server
> **Server execution:**
> Update model: $\theta_{r+1} \leftarrow \theta_r - \eta g^{(r)}$

**Output:** Optimized model parameters $\theta_R$

---

Clients locally compute the gradient $g^{(r)} = \nabla_{\theta_r} \mathcal{L}(\theta_r)$ and transmit only this gradient to the server. The server coordinates $R$ training rounds by aggregating gradients and updating global model parameters.

### 3.2 GRADIENT INVERSION ATTACK METHODOLOGY (GRADCFG)

#### 3.2.1 ATTACK MODELING

A malicious server can inversely reconstruct users' private data $\mathbf{x}_0$ and text embeddings $C_0$ when only accessing model gradients $g^{(r)}$, where noise vector $\boldsymbol{\epsilon}$ and time step $t$ remain private user information. This attack process is formalized as the following multivariate optimization problem:

$$\min_{\hat{\mathbf{x}}_0, \hat{C}_0, \hat{\boldsymbol{\epsilon}}, \hat{t}} \mathcal{D} \left( \nabla_{\theta_r} \mathcal{L}(\hat{\mathbf{x}}_0, \hat{C}_0, \hat{\boldsymbol{\epsilon}}, \hat{t}; \theta_r), g^{(r)} \right) + \eta(s) \mathcal{L}_{\mathrm{mix}}(\hat{\mathbf{x}}_0) \tag{2}$$

where $\mathcal{D}(\cdot, \cdot)$ is the gradient similarity metric function (cosine similarity is adopted in this paper), $g^{(r)} = \nabla_{\theta_r} \mathcal{L}(\mathbf{x}_0, C_0, \boldsymbol{\epsilon}, t; \theta_r)$ represents the observed true gradient, and $\nabla_{\theta_r} \mathcal{L}(\hat{\mathbf{x}}_0, \hat{C}_0, \hat{\boldsymbol{\epsilon}}, \hat{t}; \theta_r)$ is the gradient based on virtual parameters. $\mathcal{L}_{\mathrm{mix}}(\hat{\mathbf{x}}_0)$ is defined as the disentanglement regularizer, implemented by computing the mean cosine similarity of all data pairs in the reconstructed image set $\{\hat{\mathbf{x}}_0^{(1)}, \ldots, \hat{\mathbf{x}}_0^{(k)}\}$:

$$\mathcal{L}_{\mathrm{mix}}(\hat{\mathbf{x}}_0) = \frac{2}{k(k-1)} \sum_{i=1}^{k-1} \sum_{j=i+1}^{k} \frac{\langle \hat{\mathbf{x}}_0^{(i)}, \hat{\mathbf{x}}_0^{(j)} \rangle}{\|\hat{\mathbf{x}}_0^{(i)}\|_2 \cdot \|\hat{\mathbf{x}}_0^{(j)}\|_2} \tag{3}$$

The regularization strength is dynamically controlled by the iteration step scheduler $\eta(s)$, where $s$ denotes the current optimization iteration count:

$$\eta(s) = \begin{cases} \eta_{\max} & s < S_{\mathrm{switch}} \\ 0 & s \geq S_{\mathrm{switch}} \end{cases} \tag{4}$$

This scheduling strategy preserves the full regularizer $\eta_{\max} \mathcal{L}_{\mathrm{mix}}(\hat{\mathbf{x}}_0)$ during the early training phase ($s < S_{\mathrm{switch}}$) to enforce feature disentanglement and prevent feature mixing in reconstructed samples. During the later training phase ($s \geq S_{\mathrm{switch}}$), the regularizer constraint is completely removed, allowing the optimization process to focus solely on minimizing the gradient difference $\mathcal{D}(\cdot, \cdot)$. Here, $S_{\mathrm{switch}}$ is a preset iteration count threshold controlling the transition from the feature disentanglement phase to the precision optimization phase.

#### 3.2.2 QUADRUPLE COLLABORATIVE OPTIMIZATION ALGORITHM

Defining the pseudo-gradient objective function $\mathcal{G} = \mathcal{D} \left( \nabla_{\theta_r} \mathcal{L}(\hat{\mathbf{x}}_0, \hat{C}_0, \hat{\boldsymbol{\epsilon}}, \hat{t}; \theta_r), g^{(r)} \right)$ with its corresponding gradient denoted as $\nabla \mathcal{G}$, we propose a quadruple collaborative optimization framework:

**Image Reconstruction ($\mathbf{x}_0$ optimization):** To maintain latent space consistency, an initial point $\hat{X}_0 \sim \mathcal{N}(0, I)$ is sampled from image space and projected into latent space via the VAE encoder: $\hat{\mathbf{x}}_0^{(0)} = \text{VAE}(\hat{X}_0)$ Update rule: $\hat{\mathbf{x}}_0 \leftarrow \hat{\mathbf{x}}_0 - \eta_x \nabla_{\hat{\mathbf{x}}_0} \mathcal{G}$

**Text Reconstruction ($C_0$ optimization):** Initialized with empty text $\hat{P} = \phi$, projected through the text encoder: $\hat{C}_0^{(0)} = \text{Encoder}(\hat{P})$ Update rule: $\hat{C}_0 \leftarrow \hat{C}_0 - \eta_C \nabla_{\hat{C}_0} \mathcal{G}$

**Time Step Reconstruction ($t$ optimization):** To optimize the originally discrete time step $\hat{t} \in \{1, \ldots, T\}$ in continuous space, we propose a function reparameterization strategy. Addressing the discrete nature of the noise scheduler $\alpha_t$, we establish a continuous mapping through mathematical transformation. Specifically, considering the definition $\alpha_t = \prod_{i=1}^{t}(1 - \beta_i)$ where $\beta_i = f(i)$ is a predefined discrete scheduling function (e.g., linear or cosine decay), when $\beta_i \ll 1$, we utilize the natural logarithm approximation $\ln(1 - \beta_i) \approx -\beta_i$ to transform the discrete summation $\sum_{i=1}^{t} \ln(1 - \beta_i)$ into integral form:

$$\ln \alpha_t \approx -\sum_{i=1}^{t} \beta_i \approx -\int_0^t f(x)dx \tag{5}$$

This derivation yields the continuous time step representation:$\alpha(t) \approx \exp\left(-\int_0^t f(x)dx\right)$.

This continuous representation enables time step $\hat{t}$ to be updated via standard gradient descent: $\hat{t} \leftarrow \hat{t} - \eta_t \nabla_{\hat{t}} \mathcal{G}$

**Noise Reconstruction with Dynamic Reset ($\epsilon$ optimization):** Initialize $\hat{\epsilon} \sim \mathcal{N}(0, I)$. We incorporate a dynamic reset mechanism where $\hat{\epsilon}$ is randomly resampled from $\mathcal{N}(0, I)$ at fixed intervals $S_{\text{reset}}$:

$$\hat{\epsilon} \sim \mathcal{N}(0, I) \quad \text{when} \quad s \equiv 0 \pmod{S_{\text{reset}}} \tag{6}$$

This mechanism continuously searches for new solutions within the solution space, allowing other optimization variables ($\hat{\mathbf{x}}_0$, $\hat{C}_0$, and $\hat{t}$) to adapt to different solutions and achieve optimal performance. The update rule is: $\hat{\epsilon} \leftarrow \hat{\epsilon} - \eta_\epsilon \nabla_{\hat{\epsilon}} \mathcal{G}$

### 3.3 Bidirectional Sampling Enhancement Algorithm (Inv-Sam)

The preliminary reconstruction results $(\hat{\mathbf{x}}_0, \hat{C}_0) \in \mathbb{R}^{m \times B} \times \mathbb{R}^{77 \times 768}$ obtained through gradient inversion at round $r$ are mapped to natural image space via the pretrained VAE decoder: $\hat{\mathbf{X}} = \text{VAE}^{-1}(\hat{\mathbf{x}}_0)$ However, $\hat{\mathbf{X}}$ suffers from insufficient visual fidelity and missing high-frequency details. We leverage the dynamic evolution of generative capabilities during fine-tuning: compared to the initial model $\theta_0$, the final model $\theta_R$ generates images with richer training-set features under text embedding $C_0$ guidance. This paper proposes using the generative capability difference $\theta_R - \theta_0$ as an optimization prior.

Inspired by reverse diffusion and forward sampling in image editing (Miyake et al., 2024; Huang et al., 2025b), we design a text-guided latent optimization method (Algorithm 2). Using the recovered condition $\hat{C}_0$, we first perform inverse diffusion with $\theta_0$ to project $\hat{\mathbf{x}}_0$ into noise space, then execute sampling using the generative difference $\theta_R - \theta_0$ to reproject to latent space. Analysis of the inverse and sampling path relationship reveals that under path proximity (Miyake et al., 2024), their difference is approximately proportional to the model prediction difference:

$$\mathbf{x}_t^{\text{sam}} - \mathbf{x}_t^{\text{inv}} \propto \omega_{\text{sam}} \cdot \underbrace{\left(\epsilon_{\theta_R}(\mathbf{x}_{t+1}^{\text{sam}}, t+1, \hat{C}_0) - \epsilon_{\theta_0}(\mathbf{x}_{t+1}^{\text{sam}}, t+1, \hat{C}_0)\right)}_{\Delta\epsilon_\theta}$$

where $\Delta\epsilon_\theta$ quantifies the directional correction from fine-tuning. A formal proof is provided in Appendix D.

---

**Algorithm 2** Inv-Sam Optimization

---

**Input:** Initial latent state $\hat{\mathbf{x}}_0 \in \mathbb{R}^m$, reconstructed text embedding $\hat{C}_0 \in \mathbb{R}^{77 \times 768}$
Reverse step guidance factor $\omega_{\text{inv}} = 1$, sampling step guidance factor $\omega_{\text{sam}}$
Noise schedule $\{\bar{\alpha}_t\}_{t=0}^T$
Fine-tuned model parameters $\theta_R$, initial model parameters $\theta_0$
**Phase I: Inverse Diffusion**     $\mathbf{x}_0^{\text{inv}} \leftarrow \hat{\mathbf{x}}_0$
**for** $t = 0$ **to** $T - 1$ **do**

$\quad$ $\tilde{\epsilon}^{\text{inv}} \leftarrow \epsilon_{\theta_0}(\mathbf{x}_t^{\text{inv}}, t, \hat{C}_0)$

$\quad$ $\mathbf{x}_{t+1}^{\text{inv}} \leftarrow \sqrt{\bar{\alpha}_{t+1}} \left( \frac{\mathbf{x}_t^{\text{inv}} - \sqrt{1 - \bar{\alpha}_t}\tilde{\epsilon}^{\text{inv}}}{\sqrt{\bar{\alpha}_t}} \right) + \sqrt{1 - \bar{\alpha}_{t+1}}\tilde{\epsilon}^{\text{inv}}$

**Phase II: Conditional Sampling**     $\mathbf{x}_T^{\text{sam}} \leftarrow \mathbf{x}_T^{\text{inv}}$
**for** $t = T - 1$ **to** $0$ **do**

$\quad$ $\epsilon_{\text{empty}} \leftarrow \epsilon_{\theta_0}(\mathbf{x}_{t+1}^{\text{sam}}, t+1, \hat{C}_0)$

$\quad$ $\epsilon_{\text{text}} \leftarrow \epsilon_{\theta_R}(\mathbf{x}_{t+1}^{\text{sam}}, t+1, \hat{C}_0)$

$\quad$ $\tilde{\epsilon}^{\text{sam}} \leftarrow \epsilon_{\text{empty}} + \omega_{\text{sam}}(\epsilon_{\text{text}} - \epsilon_{\text{empty}})$

$\quad$ $\mathbf{x}_t^{\text{sam}} \leftarrow \sqrt{\bar{\alpha}_t} \left( \frac{\mathbf{x}_{t+1}^{\text{sam}} - \sqrt{1 - \bar{\alpha}_{t+1}}\tilde{\epsilon}^{\text{sam}}}{\sqrt{\bar{\alpha}_{t+1}}} \right) + \sqrt{1 - \bar{\alpha}_t}\tilde{\epsilon}^{\text{sam}}$

**Output:** Optimized latent state $\hat{\mathbf{x}}_0^{\text{opt}} \leftarrow \mathbf{x}_0^{\text{sam}}$

---

## 4 EXPERIMENTAL EVALUATION

### 4.1 EXPERIMENTAL SETUP

Experiments utilize the standard DREAMBOOTH (Ruiz et al., 2023) training dataset containing image samples at $512 \times 512$ resolution. All models are fine-tuned using the TinySD (Kim et al., 2023) framework to ensure parameter-efficient optimization. Two experimental scenarios are designed:

**Gradient inversion for generic text prompt fine-tuning**: Constructs a category-uniform scenario where images of similar objects (e.g., backpacks in different environments) are fine-tuned using unified text prompts (e.g., "backpack").

**Gradient inversion for specific text prompt fine-tuning**: Constructs a fine-grained control scenario where each image is paired with a dedicated granular prompt (e.g., "a red backpack on a mountain trail") to evaluate reconstruction performance under complex text conditions.

To the best of our knowledge, this is the first work to study gradient inversion attacks on CFG-based diffusion models. Existing approaches are unable to jointly recover both images and text prompts from gradients under this setting. While no directly comparable baselines currently exist, we still make an effort to construct several reasonable baseline variants in Appendix I. These baselines allow us to carefully analyze and contrast our method with alternative designs. Therefore, our main evaluation compares our full pipeline (GradCFG + Inv-Sam) with these constructed baselines and with its own ablated variants to isolate the contribution of each component.

### 4.2 EVALUATION METRICS

To comprehensively assess reconstruction quality, we employ both quantitative and qualitative evaluation frameworks. Image reconstruction quality is measured using three complementary metrics: Peak Signal-to-Noise Ratio (PSNR) (Wang et al., 2004) for pixel-level fidelity assessment, Learned Perceptual Image Patch Similarity (LPIPS) (Zhang et al., 2018) for human visual perception similarity (with lower values indicating better performance), and the Structural Similarity Index Measure (SSIM) (Wang et al., 2004) for assessing the perceptual quality related to structural information, luminance, and contrast.

For semantic recovery evaluation of generic prompts, we implement a dual-modality analysis framework that computes cosine similarity between reconstructed and original prompts in embedding space to quantify semantic consistency, while also generating images guided by original and reconstructed prompts under identical initial noise conditions using the TinySD model to compute PSNR between paired images as an indirect measure of semantic recovery effectiveness.

## 5 RESULTS

### 5.1 IMAGE RECONSTRUCTION ANALYSIS

We conducted gradient inversion experiments under two distinct fine-tuning scenarios: generic text prompts and detailed text prompts. The reconstruction process employed a two-stage approach: initial image reconstruction was performed using the GradCFG method, followed by detail enhancement via the Inv-Sam algorithm.

Table 1 presents comparative reconstruction results under different text prompts. Experimental results demonstrate that baseline gradient inversion achieves preliminary reconstruction for both prompt types, with comparable quantitative metrics (structural similarity SSIM∼0.12, peak signal-to-noise ratio PSNR∼10.6 dB, perceptual similarity LPIPS∼0.78). Inv-Sam optimization significantly enhances reconstruction quality: under generic prompts, SSIM increases by 77% to 0.219 while LPIPS decreases by 24% to 0.591. For specific prompts, SSIM improvement reaches 55% with a 20% reduction in LPIPS.

Table 1: Experimental results of GradCFG and Inv-Sam under different prompt settings (Values in parentheses indicate metric changes after applying Inv-Sam)

| Setting | GradCFG | | | + Inv-Sam | | |
|---|---|---|---|---|---|---|
| | SSIM ↑ | PSNR ↑ | LPIPS ↓ | SSIM ↑ | PSNR ↑ | LPIPS ↓ |
| Generic prompts | 0.1240 | 10.60 | 0.7778 | 0.2189 (+77%) | 11.65 (+10%) | 0.5911 (-24%) |
| Specific prompts | 0.1213 | 10.65 | 0.7789 | 0.1875 (+55%) | 11.51 (+8%) | 0.6226 (-20%) |

Visual results are presented in Figure 2 (generic prompts) and Figure 3 (specific prompts). The gradient inversion stage effectively captures object contours and base colors, establishing structural frameworks. Subsequently, Inv-Sam refinement enhances textural details, enabling reconstruction of fine-grained features.

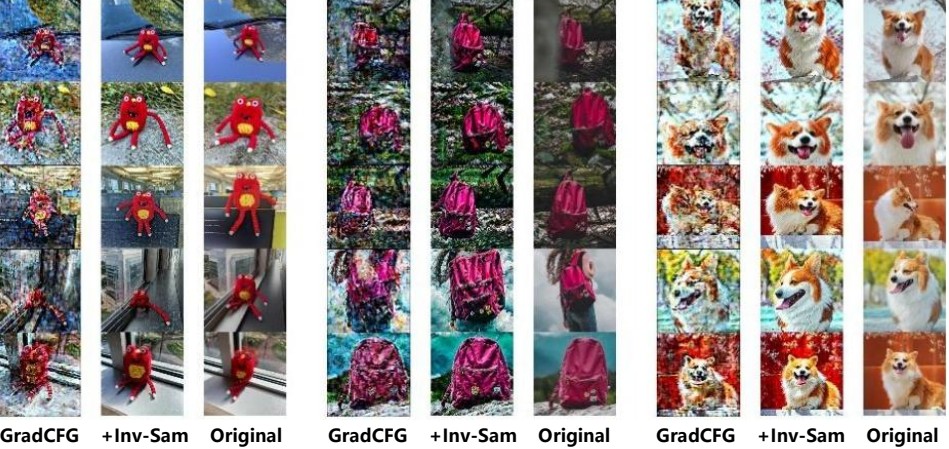

**GradCFG   +Inv-Sam   Original          GradCFG   +Inv-Sam   Original          GradCFG   +Inv-Sam   Original**

Figure 2: Reconstruction workflow under generic prompts: GradCFG results (left), Inv-Sam optimized results (middle), Original picture (right)

Comprehensive analysis confirms the effective synergy: gradient inversion recovers structural frameworks, while Inv-Sam enhances texture details. This mechanism achieves greater metric improvements under generic prompts while maintaining robustness for specific prompt scenarios.

### 5.2 TEXT EMBEDDING RECOVERY ANALYSIS

In the fine-tuning scenario with generic text prompts, we conducted a systematic analysis of text encoding embeddings. The evaluation procedure consisted of three sequential steps: First, we computed the cosine similarity between the recovered text embeddings and the original text embeddings.

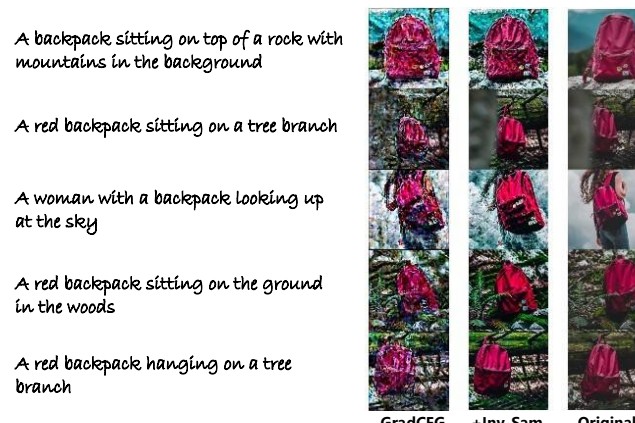

A backpack sitting on top of a rock with mountains in the background

A red backpack sitting on a tree branch

A woman with a backpack looking up at the sky

A red backpack sitting on the ground in the woods

A red backpack hanging on a tree branch

GradCFG   +Inv-Sam   Original

Figure 3: Reconstruction results for backpack category images under various scenes.

Subsequently, both the recovered and original embeddings were separately employed as conditional guidance inputs to a pre-trained CFG model to generate corresponding images. Finally, we quantified the similarity between these generated image pairs using PSNR, thereby validating the semantic consistency of the recovered embeddings.

Quantitative results presented in Table 2 demonstrate that the gradient inversion method effectively recovers generic text embeddings while maintaining high semantic fidelity (cosine similarity: 0.7953; PSNR: 16.28 dB). Visual comparisons in Figure 4 further confirm that images generated from recovered embeddings exhibit strong semantic alignment with those produced from original embeddings, validating the method's effectiveness in preserving semantic information integrity.

Table 2: Recovery Performance for Generic Embeddings

| Similarity | PSNR |
|---|---|
| 0.7953 | 16.28 |

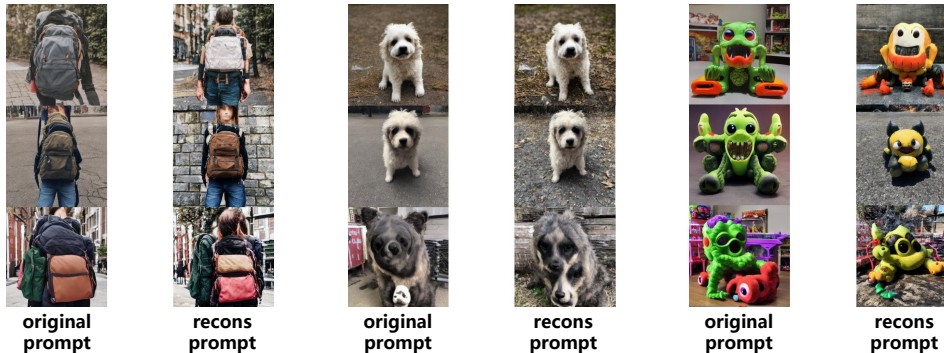

| original prompt | recons prompt | original prompt | recons prompt | original prompt | recons prompt |

Figure 4: Generated image comparison based on text embeddings (left to right: backpack, dog, monster toy). For each group: left image generated from original prompt embedding, right image from reconstructed text embedding.

### 5.3 IMPACT OF $\epsilon$ RESET CYCLE ON RECONSTRUCTION QUALITY

This experiment systematically investigates the influence of different $\epsilon$ reset cycles on image reconstruction quality in gradient inversion attacks. As shown in Figure 5, under a fixed total optimization budget of 4000 iterations, we evaluated reconstruction performance at reset cycles of 1, 10, 100, 1000, and 4000 (no reset).Experimental results reveal significant performance variations based on reset cycle selection. When the reset cycle is too short (e.g., 1 or 10), the $\epsilon$ parameter cannot sufficiently optimize to convergence regions, causing estimation bias in latent variables. This manifests as reconstructed images with clear pixel-level details but noticeable structural misalignments and semantic inconsistencies. Conversely, excessively long reset cycles (e.g., 1000) or no reset (4000) prevent

adequate exploration of diversity within the $\epsilon$ solution space, causing optimization to stagnate in local minima. This results in reconstructed images with blurred details and lacking texture features.

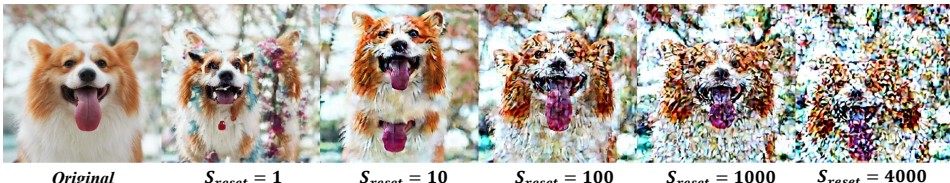

*Original*     $S_{reset} = 1$     $S_{reset} = 10$     $S_{reset} = 100$     $S_{reset} = 1000$     $S_{reset} = 4000$

Figure 5: Impact of $\epsilon$ reset cycles on image reconstruction quality. From left to right: reset cycles = 1, 10, 100, 1000, 4000 (no reset). Experimental results demonstrate that a moderate reset cycle (100) optimally balances optimization stability and solution space exploration capability.

In conclusion, the $\epsilon$ reset cycle requires careful balancing between optimization stability and solution space exploration capacity. Our experimental findings indicate that a reset cycle of 100 achieves optimal reconstruction quality, ensuring sufficient $\epsilon$ parameter optimization while maintaining effective solution space exploration.

### 5.4 ANALYSIS OF NON-UNIQUENESS IN $\epsilon$ SOLUTIONS

In this experiment, we use both the SD 1.4 model and the Tiny-SD model. For each model, we select 100 different random initializations of the noise $\epsilon$ and independently optimize them under known $\mathbf{x}_0$, $C_0$, and $t$. During optimization, we track two quantities: (1) the average similarity between the recovered noise $\hat{\epsilon}$ and the ground-truth noise $\epsilon$, and (2) the average similarity between the simulated gradient $g$ and the original gradient $g_0$. Figure 6 shows the evolution of these two curves for both models.

We observe that, for both SD 1.4 and Tiny-SD, the simulated gradient $g$ can become highly aligned with $g_0$ (similarity $> 0.9$), while the similarity between $\hat{\epsilon}$ and $\epsilon$ remains low (around 0.2). This indicates that strict noise-level alignment is not required to achieve strong gradient alignment: multiple distinct $\epsilon$ configurations can induce nearly identical gradients, implying non-uniqueness of feasible $\epsilon$. Therefore, our reconstruction method is designed so that the recovered variables remain valid across many possible $\epsilon$ solutions. Concretely, we periodically reinitialize $\epsilon$ during optimization, encouraging the remaining variables to satisfy gradient alignment under diverse, plausible noise realizations.

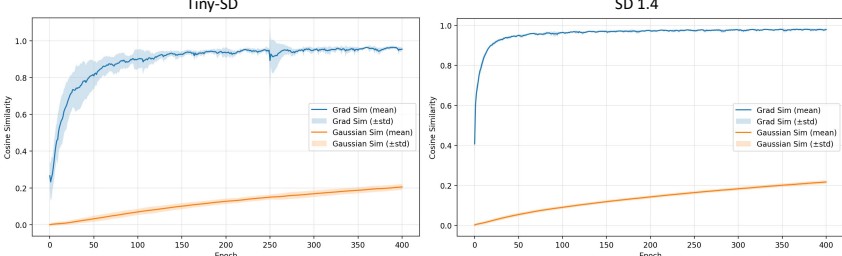

Figure 6: Optimization of randomly sampled noise $\epsilon$ under fixed $\mathbf{x}_0$, $C_0$, and $t$, while monitoring gradient alignment and noise similarity relative to the original data. Left: Tiny-SD model; right: SD 1.4 model.

## 6 CONCLUSION

This study introduces a novel gradient inversion technique for reconstructing high-resolution images and their corresponding text prompts during CFG-based model training. The proposed GradCFG method enables, for the first time, simultaneous recovery of both training images and associated text prompts, overcoming a key limitation of prior approaches that struggle with joint visual and semantic reconstruction. An enhancement module, Inv-Sam, leverages the generative gap between fine-tuned and initial models as prior knowledge, substantially improving image quality and semantic accuracy. Experiments conducted under a DREAMBOOTH-like fine-tuning setup using TinySD models demonstrate high-fidelity reconstruction of $512 \times 512$ complex scenes and accurate text recovery. The method performs robustly across both generic and specific prompts, regardless of complexity.

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

## A    ETHICS STATEMENT

This work investigates potential privacy vulnerabilities in classifier-free guidance (CFG) diffusion models during personalized fine-tuning. We strictly adhere to the ICLR Code of Ethics and are fully aware of the dual-use implications of our research. To mitigate potential risks, all experiments exclusively utilize publicly available benchmark datasets (DREAMBOOTH), ensuring no private or sensitive data is involved. The fundamental objective of this research is to raise community awareness about privacy leakage threats in distributed learning scenarios, with the ultimate goal of contributing to more secure federated learning frameworks. We emphasize that the defensive value of understanding these vulnerabilities significantly outweighs any offensive potential. While our method demonstrates reconstruction capabilities, we strongly oppose any malicious application of this technique and believe transparent analysis of such attack vectors is essential for developing robust defenses.

## B    REPRODUCIBILITY STATEMENT

To ensure the reproducibility of this research, comprehensive efforts have been made to provide complete implementation details and resources. Full implementation code, pretrained model weights, and evaluation scripts are available in the supplementary materials. The core methodologies (GradCFG and Inv-Sam algorithms) are thoroughly described in Sec. 3, including optimization objectives, loss functions, and key reparameterization strategies. Complete hyperparameter configurations (learning rates, reset period $S_{reset}$, etc.) are documented in Appendix. E.

## C    ON THE USE OF LARGE LANGUAGE MODELS

Large language models (LLMs) were employed solely for writing assistance, including surface-level text editing (grammar correction, clarity improvement), document formatting, and experimental code comment generation. LLMs did not contribute to originating research ideas, claims, or conclusions. The authors take full responsibility for all intellectual content. All LLM-assisted text was carefully reviewed and rewritten by the authors to ensure accurate expression of the research.

## D    PROOF OF INV-SAM

**Why path proximity implies proportional correction.** We formalize this relationship as the following Prop. D.1 and provide a dimension-free proof based on linear operator analysis.

**Proposition D.1** (Proportionality of sampling-inverse path difference to model prediction difference)**.** *Consider the DDIM framework with linear update function $g_t(\mathbf{x}, \epsilon) = A_t\mathbf{x} + B_t\epsilon$, where*

$$A_t = \frac{\sqrt{\bar{\alpha}_t}}{\sqrt{\bar{\alpha}_{t+1}}}, \quad B_t = \sqrt{1 - \bar{\alpha}_t} - \frac{\sqrt{\bar{\alpha}_t}\sqrt{1 - \bar{\alpha}_{t+1}}}{\sqrt{\bar{\alpha}_{t+1}}}. \tag{7}$$

*Define the path difference $\delta_t = \mathbf{x}_t^{sam} - \mathbf{x}_t^{inv}$, model prediction difference $\Delta\epsilon_\theta = \epsilon_{\theta_R}(\mathbf{x}_{t+1}^{sam}, t+1, \hat{C}_0) - \epsilon_{\theta_0}(\mathbf{x}_{t+1}^{sam}, t+1, \hat{C}_0)$, and sampling noise $\tilde{\epsilon}^{sam} = \epsilon_{\theta_0}(\mathbf{x}_{t+1}^{sam}, t+1, \hat{C}_0) + \omega_{sam}\Delta\epsilon_\theta$. Under the path proximity assumption $\mathbf{x}_{t+1}^{sam} \approx \mathbf{x}_{t+1}^{inv}$ (implying $\delta_{t+1} \approx 0$ and similar denoising outputs), we have*

$$\delta_t \approx B_t\omega_{sam}\Delta\epsilon_\theta. \tag{8}$$

*In particular, the path difference is proportional to the guidance-weighted model prediction difference:*

$$\mathbf{x}_t^{sam} - \mathbf{x}_t^{inv} \propto \omega_{sam} \cdot \Delta\epsilon_\theta \tag{9}$$

*with proportionality constant $B_t$ depending only on the noise schedule.*

*Proof of Prop. D.1.* Starting from the definition of the path difference and substituting the update operations:

$$\delta_t = g_t(\mathbf{x}_{t+1}^{sam}, \tilde{\epsilon}^{sam}) - g_t(\mathbf{x}_{t+1}^{inv}, \epsilon_{\theta_0}(\mathbf{x}_{t+1}^{inv}, t+1, \hat{C}_0)) \tag{10}$$

Exploiting the linearity of the update function $g_t$:

$$\delta_t = A_t(\mathbf{x}_{t+1}^{\text{sam}} - \mathbf{x}_{t+1}^{\text{inv}}) + B_t(\tilde{\epsilon}^{\text{sam}} - \epsilon_{\theta_0}(\mathbf{x}_{t+1}^{\text{inv}}, t+1, \hat{C}_0)) \tag{11}$$

Substituting $\delta_{t+1}$ and applying the path proximity assumption ($\delta_{t+1} \approx 0$):

$$\delta_t \approx B_t(\tilde{\epsilon}^{\text{sam}} - \epsilon_{\theta_0}(\mathbf{x}_{t+1}^{\text{inv}}, t+1, \hat{C}_0)) \tag{12}$$

Expanding the sampling noise $\tilde{\epsilon}^{\text{sam}}$:

$$\delta_t \approx B_t \left[ \epsilon_{\theta_0}(\mathbf{x}_{t+1}^{\text{sam}}, t+1, \hat{C}_0) + \omega_{\text{sam}}\Delta\epsilon_\theta - \epsilon_{\theta_0}(\mathbf{x}_{t+1}^{\text{inv}}, t+1, \hat{C}_0) \right] \tag{13}$$

By path proximity, the denoising outputs are similar:

$$\epsilon_{\theta_0}(\mathbf{x}_{t+1}^{\text{sam}}, t+1, \hat{C}_0) \approx \epsilon_{\theta_0}(\mathbf{x}_{t+1}^{\text{inv}}, t+1, \hat{C}_0) \tag{14}$$

Thus the terms cancel, yielding the final result:

$$\delta_t \approx B_t \omega_{\text{sam}} \Delta\epsilon_\theta \tag{15}$$

This establishes the proportionality with schedule-dependent constant $B_t$. $\square$

The interpretation reveals that $B_t$ represents the time-dependent scaling from the noise schedule, while $\omega_{\text{sam}}$ directly controls the amplification of model corrections. This proportionality demonstrates that our latent optimization algorithm effectively translates fine-tuning improvements into controlled path deviations, maintaining the delicate balance between faithfulness to the input and incorporation of desired model enhancements.

## E  FURTHER EXPERIMENTAL DETAILS

All experiments were conducted using the pre-trained TinySD model as the base architecture (a lightweight version of Stable Diffusion with $3 \times 10^8$ parameters) on NVIDIA A800 80GB PCIe GPU platforms. The optimization process employed the Adam optimizer ($\beta_1 = 0.8, \beta_2 = 0.9$) with the following learning rate configuration: 0.1 for image latent variables $\hat{\mathbf{x}}_0$, 0.001 for text embeddings $\hat{C}_0$, and 0.1 for both noise parameters $\hat{\epsilon}$ and timesteps $\hat{t}$. The total number of iterations was fixed at 4000, with periodic reset of noise parameters $\hat{\epsilon}$ implemented every 100 steps. To prevent vanishing gradients during optimization, we constrained the temporal range of $\hat{t}$ to values between 400 and 600 throughout the reconstruction process. A default batch size of $B = 5$ was used throughout, with gradient alignment measured using a cosine similarity-based distance function $\mathcal{D}(\cdot) = 1 - \cos(\cdot)$. The feature decoupling regularization term $\mathcal{S}(\cdot)$ was activated during the first 100 iterations to enhance initial convergence stability. All experiments were performed under identical configuration environments to ensure result comparability and reproducibility.

## F  DETAILED EXPERIMENTAL RESULTS OF GRADCFG

### F.1  DETAILED RESULTS FOR FINETUNING EXPERIMENTS WITH GENERIC TEXT PROMPTS

This section provides the complete experimental results for the finetuning experiments using generic text prompts, serving as supplementary data to Section 5.1. Table 3 presents the comprehensive performance comparison between baseline GradCFG and our Inv-Sam enhanced approach across all object categories.

Table 4 provides the detailed semantic recovery metrics, including embedding similarity scores and image similarity measurements for each category.

Table 3: GradCFG results using generic text prompts for fine-tuning.

| Category | GradCFG | | | + Inv-Sam | | |
|---|---|---|---|---|---|---|
| | SSIM ↑ | PSNR ↑ | LPIPS ↓ | SSIM ↑ | PSNR ↑ | LPIPS ↓ |
| Backpack | 0.1267 | 10.84 | 0.7222 | 0.2441 | 11.97 | 0.5672 |
| Can | 0.1706 | 10.10 | 0.8073 | 0.3984 | 11.70 | 0.5813 |
| Candle | 0.0684 | 10.43 | 0.8096 | 0.1251 | 12.16 | 0.5640 |
| Cat | 0.1710 | 12.20 | 0.6745 | 0.2259 | 13.76 | 0.5321 |
| Sneaker | 0.0555 | 9.97 | 0.9142 | 0.1024 | 11.48 | 0.6376 |
| Dog | 0.2095 | 10.98 | 0.7446 | 0.3271 | 12.53 | 0.5690 |
| Monster Toy | 0.1298 | 10.43 | 0.7705 | 0.2706 | 11.78 | 0.5770 |
| Robot Toy | 0.0692 | 9.28 | 0.7801 | 0.1165 | 10.20 | 0.6102 |
| Race Car | 0.1170 | 10.15 | 0.7702 | 0.1623 | 10.87 | 0.6635 |
| **Average** | 0.1240 | 10.60 | 0.7778 | 0.2189 | 11.65 | 0.5911 |

Table 4: Evaluation of semantic recovery for text prompts

| Category | Similarity | PSNR |
|---|---|---|
| Backpack | 0.7538 | 14.94 |
| Can | 0.9228 | 16.69 |
| Candle | 0.8000 | 17.52 |
| Cat | 0.8457 | 18.08 |
| Sneaker | 0.7010 | 13.06 |
| Dog | 0.8766 | 19.49 |
| Monster Toy | 0.7330 | 13.10 |
| Robot Toy | 0.7368 | 15.30 |
| Race Car | 0.7878 | 17.32 |
| **Avg.** | **0.7953** | **16.28** |

## F.2 DETAILED RESULTS FOR FINETUNING EXPERIMENTS WITH SPECIFIC TEXT PROMPTS

This section provides the complete experimental results for finetuning experiments using specific text prompts, serving as supplementary data to Section 5.1. Table 5 presents the comprehensive performance comparison between baseline GradCFG and our Inv-Sam enhanced approach under complex textual conditions.

Table 5: GradCFG results using specific text prompts for fine-tuning

| Category | GradCFG | | | + Inv-Sam | | |
|---|---|---|---|---|---|---|
| | SSIM ↑ | PSNR ↑ | LPIPS ↓ | SSIM ↑ | PSNR ↑ | LPIPS ↓ |
| Backpack | 0.1381 | 10.81 | 0.7252 | 0.2277 | 11.52 | 0.6253 |
| Can | 0.1626 | 10.03 | 0.8099 | 0.2037 | 10.62 | 0.7403 |
| Candle | 0.0743 | 10.28 | 0.8037 | 0.1222 | 11.73 | 0.5836 |
| Cat | 0.1553 | 11.78 | 0.7028 | 0.1840 | 13.28 | 0.5539 |
| Sneaker | 0.0627 | 10.35 | 0.9303 | 0.1272 | 12.43 | 0.6155 |
| Dog | 0.1861 | 10.43 | 0.7619 | 0.2583 | 11.33 | 0.6520 |
| Monster Toy | 0.1134 | 10.20 | 0.7913 | 0.2346 | 11.28 | 0.6251 |
| Robot Toy | 0.1142 | 10.61 | 0.7786 | 0.2230 | 11.36 | 0.6168 |
| Race Car | 0.0836 | 10.38 | 0.7546 | 0.1215 | 11.07 | 0.5652 |
| **Avg.** | **0.1213** | **10.65** | **0.7789** | **0.1875** | **11.51** | **0.6226** |

For complex text recovery tasks, this experiment first analyzes the text reconstruction performance using backpack-related prompts as a detailed case study, followed by a comprehensive analysis across all text categories. It is important to note that complete reconstruction of all textual information is not our primary objective, as full recovery of complex semantic content presents significant challenges.

Instead, we focus on evaluating the improvement in similarity between recovered text embeddings and original data compared to null text embeddings. As demonstrated in Table 6, our method achieves significant improvements in this measured similarity for specific prompt examples. Table 7 further presents the overall performance across all categories, showing that our approach substantially enhances text embedding recovery quality by this metric, effectively demonstrating the utility of our method without requiring complete semantic reconstruction.

Table 6: Semantic similarity comparison between reconstructed text embeddings and original prompts

| Original Prompt | Null Text Sim. | Recon. Sim. | Improv. (%) |
|---|---|---|---|
| A backpack sitting on top of a rock with mountains in the background | 0.244 | 0.375 | 53.7 |
| A red backpack sitting on a tree branch | 0.269 | 0.477 | 77.2 |
| A woman with a backpack looking up at the sky | 0.302 | 0.351 | 16.5 |
| A red backpack sitting on the ground in the woods | 0.326 | 0.480 | 47.5 |
| A red backpack hanging on a tree branch | 0.274 | 0.463 | 69.1 |
| **Avg.** | **0.283** | **0.429** | **52.8** |

Table 7: Overall text embedding recovery performance

| Null Text Sim. | Recon. Sim. | Improv. (%) |
|---|---|---|
| 0.3178 | 0.4536 | 42.7 |

## G  VISUAL COMPARISON OF INV-SAM GUIDANCE SCALES

In this experiment, we systematically investigate the optimization effects of different guidance scales $\omega_{sam}$ in the Inv-Sam Algorithm. 2 on preliminary reconstruction results. As shown in Figure 7, the visual quality of reconstructed images progressively improves with increasing $\omega_{sam}$ values. Notably, when $\omega_{sam} = 0$, the reconstruction maintains the structural integrity and content fidelity of the initial recovery without introducing distortion or artifacts. This demonstrates that our algorithm can effectively enhance local details while preserving the fundamental framework of preliminary reconstructions, highlighting its robustness and controllability during detail refinement.

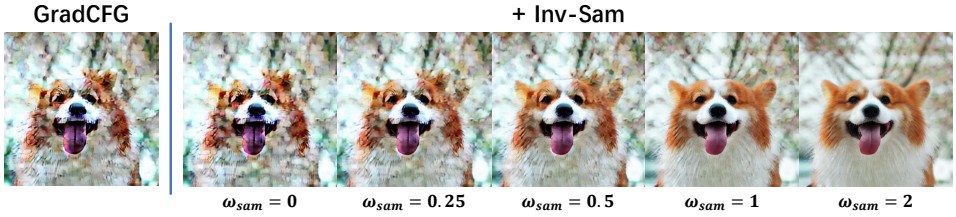

GradCFG        + Inv-Sam

$\omega_{sam} = 0$    $\omega_{sam} = 0.25$    $\omega_{sam} = 0.5$    $\omega_{sam} = 1$    $\omega_{sam} = 2$

Figure 7: Visual comparison of reconstructed images using different $\omega_{sam}$ values. From left to right: Original image, GradCFG baseline and Inv-Sam with $\omega_{sam} = 0, 0.25, 0.5, 1.0, 2.0$. Higher guidance scales generally produce sharper details and better semantic alignment with the original prompt.

## H  GENERALIZATION EXPERIMENTS ON LARGER MODELS

To validate the generalization of GradCFG to larger diffusion models, we evaluate the method on the `sd1.4` model. For each object category, we set the batch size to 2 and apply GradCFG and Inv-Sam

to recover both images and text, thereby assessing the performance of our approach on a large-scale CFG model.

## H.1 IMAGE RECONSTRUCTION

We apply GradCFG and Inv-Sam to reconstruct images and report SSIM, PSNR and LPIPS as quantitative metrics. Table 8 presents the per-category metric results; Figure 8 shows two-stage reconstruction examples (recovered results alongside the ground-truth) for the categories dog and robot toy. Overall, GradCFG achieves comparable quantitative scores and satisfactory visual quality on `sd1.4`, indicating robustness of the method across CFG models of different scales.

Table 8: Image reconstruction performance of GradCFG and Inv-Sam on `sd1.4` (SSIM / PSNR / LPIPS)

| Category | GradCFG | | | + Inv-Sam | | |
|---|---|---|---|---|---|---|
| | SSIM ↑ | PSNR ↑ | LPIPS ↓ | SSIM ↑ | PSNR ↑ | LPIPS ↓ |
| Backpack | 0.1568 | 10.120 | 0.7645 | 0.4511 | 12.110 | 0.5011 |
| Cat | 0.1219 | 10.860 | 0.6761 | 0.2367 | 11.590 | 0.5165 |
| Dog | 0.2137 | 11.180 | 0.6995 | 0.2652 | 11.250 | 0.6071 |
| Monster Toy | 0.1388 | 10.550 | 0.7541 | 0.3440 | 13.320 | 0.5120 |
| Robot Toy | 0.0821 | 9.120 | 0.7400 | 0.1415 | 10.370 | 0.5623 |
| **Avg.** | **0.1427** | **10.366** | **0.7268** | **0.2877** | **11.728** | **0.5398** |

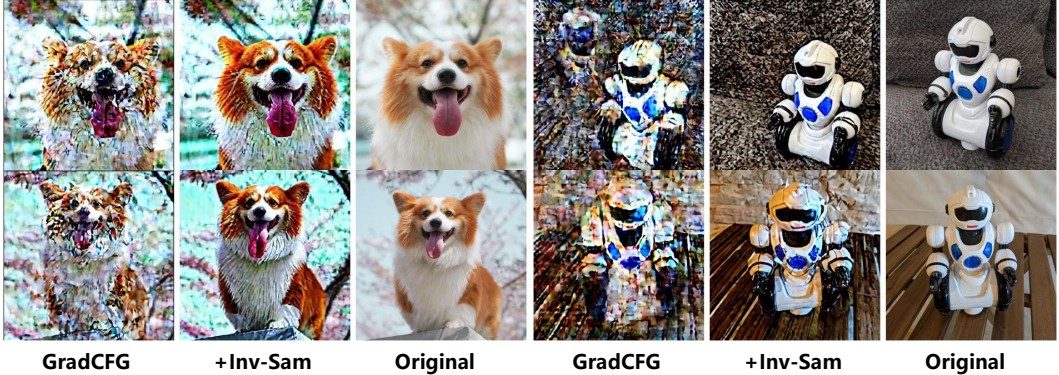

GradCFG     +Inv-Sam     Original     GradCFG     +Inv-Sam     Original

Figure 8: Example image reconstructions on `sd1.4` using GradCFG and GradCFG + Inv-Sam (categories: dog, robot toy). From left to right: ground-truth, GradCFG reconstruction, GradCFG + Inv-Sam refinement.

## H.2 TEXT RECONSTRUCTION

To evaluate semantic fidelity of the recovered text, we first compute cosine similarity between embeddings of recovered and ground-truth prompts. Then we use both recovered and ground-truth text embeddings to condition `sd1.4` and generate images; the PSNR between generated images serves as a proxy for semantic consistency. Table 9 reports per-category embedding similarity and mean PSNR; Figure 9 shows sample generations conditioned on recovered vs. ground-truth embeddings for dog and robot toy. Results indicate that GradCFG recovers meaningful textual semantics on `sd1.4` and preserves semantic consistency in downstream generation.

Table 9: Embedding similarity and generated-image PSNR comparison per category on `sd1.4`.

| Category | Similarity | PSNR |
|---|---|---|
| Backpack | 0.6538 | 13.47 |
| Cat | 0.8174 | 15.60 |
| Dog | 0.8641 | 17.20 |
| Monster Toy | 0.6311 | 12.45 |
| Robot Toy | 0.6201 | 15.34 |
| **Avg.** | **0.7173** | **14.012** |

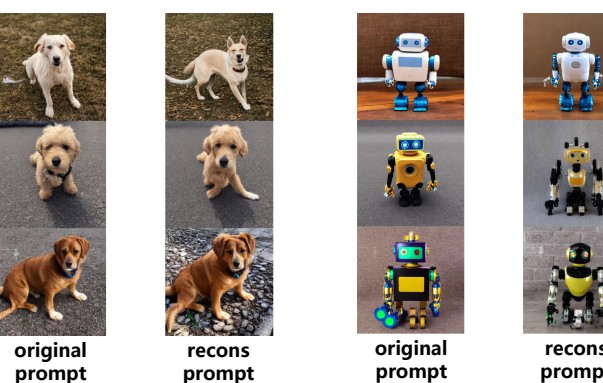

Figure 9: Sample generations conditioned on recovered vs. ground-truth text embeddings on `sd1.4` (categories: dog, robot toy).

In summary, these experiments demonstrate that GradCFG can recover both image details and meaningful textual semantics on the larger `sd1.4` model, supporting the method's transferability and robustness across CFG model scales.

## I    COMPARISON WITH EXISTING METHODS

Implementing gradient inversion attacks on CFG (Classifier-Free Guidance) models is particularly challenging because these models are larger in scale and their training involves multimodal data (image and text). Consequently, there is currently no prior work that directly provides a gradient inversion method operating on CFG models. We identified a line of diffusion-model inversion methods designed for prompt-free settings, such as GIDM, but their original designs do not support recovery of text prompts and thus cannot be directly applied to CFG text recovery.

To construct a fair and informative baseline, we adapt GIDM to our setting by explicitly supplying the ground-truth prompt as a prior during image reconstruction (i.e., we inject the textual information into GIDM in the experiments) to approximate its upper-bound performance when prompt information is available. Table 10 reports GIDM's per-category reconstruction metrics under the "known prompt" condition; Table 11 summarizes the overall comparison between GIDM (given prompt) and GradCFG (no prompt); Figure 10 provides a visual comparison of the two methods' reconstructions.

Table 10: Image reconstruction results of GIDM under known-prompt condition (per-category).

| Category | SSIM ↑ | PSNR ↑ | LPIPS ↓ |
|---|---|---|---|
| Backpack | 0.0559 | 8.547 | 0.8276 |
| Can | 0.0658 | 8.029 | 0.9089 |
| Candle | 0.0461 | 7.959 | 0.8365 |
| Cat | 0.0435 | 8.012 | 0.8534 |
| Sneaker | 0.0377 | 8.610 | 0.9566 |
| Dog | 0.1245 | 9.012 | 0.8112 |
| Monster Toy | 0.0506 | 7.930 | 0.8485 |
| Robot Toy | 0.0356 | 8.151 | 0.8228 |
| Race Car | 0.0477 | 8.213 | 0.8539 |
| **Avg.** | **0.0564** | **8.273** | **0.8566** |

Table 11: Overall comparison between GIDM (given prompt) and GradCFG (no prompt) on image reconstruction.

| Method | SSIM ↑ | PSNR ↑ | LPIPS ↓ |
|---|---|---|---|
| GIDM (given prompt) | 0.0564 | 8.273 | 0.8566 |
| GradCFG (no prompt) | 0.1240 | 10.60 | 0.7778 |

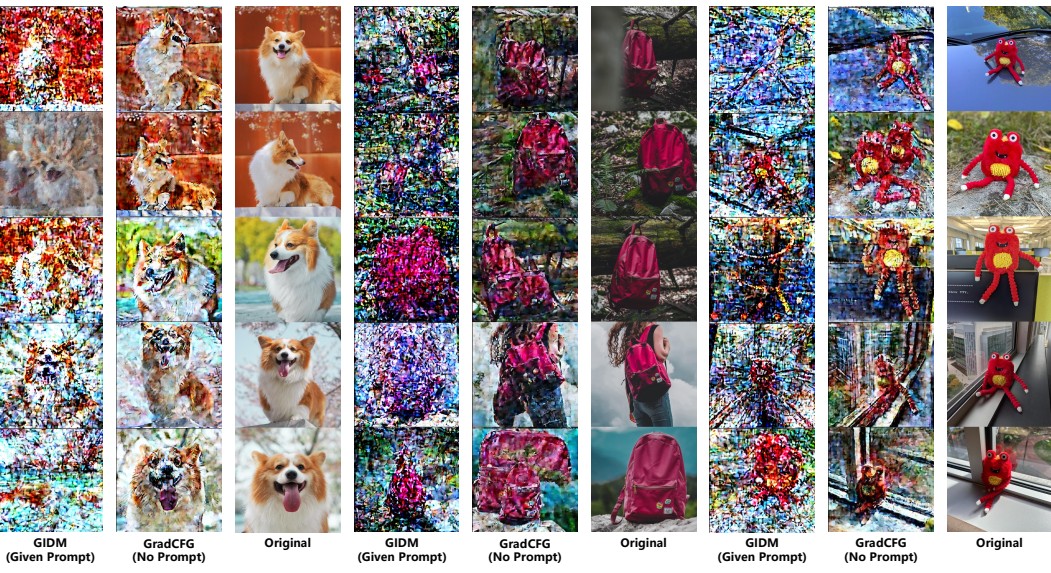

Figure 10: Visual comparison between GIDM and GradCFG (reconstructions vs. ground-truth).

From the quantitative and visual comparisons we observe that, even when GIDM is given the true prompt (a favorable condition for the baseline), GradCFG—operating without access to the prompt—still substantially outperforms GIDM across all three metrics (SSIM increases from 0.0564 to 0.1240, ≈ +120%; PSNR increases from 8.273 to 10.60, ≈ +28%; LPIPS decreases from 0.8566 to 0.7778, ≈ −10%). This comparison highlights two points: (i) directly ported prompt-free diffusion inversion methods have a limited performance ceiling on CFG multimodal tasks, and (ii) GradCFG demonstrates stronger recovery capability and robustness when dealing with higher uncertainty and multimodal coupling.

## J  ABLATION STUDY ON $L_{mix}$

To investigate the role of $L_{mix}$ in suppressing feature mixing and promoting feature disentanglement, we perform an ablation study on its activation schedule. We introduce the indicator $S_{switch}$, defined as the iteration after which $L_{mix}$ is no longer included in the loss (i.e., $L_{mix}$ is applied only during the first $S_{switch}$ optimization steps). Table 12 reports GradCFG's quantitative performance under different $S_{switch}$ settings, and Figure 11 provides the corresponding visual comparisons.

Table 12: Effect of different $S_{switch}$ settings on reconstruction performance

| $S_{switch}$ | SSIM ↑ | PSNR ↑ | LPIPS ↓ |
|---|---|---|---|
| 0 | 0.2486 | 11.210 | 0.7021 |
| 100 | 0.2655 | 11.280 | 0.6877 |
| 1000 | 0.2639 | 11.380 | 0.6940 |
| 4000 | 0.2522 | 11.170 | 0.6828 |

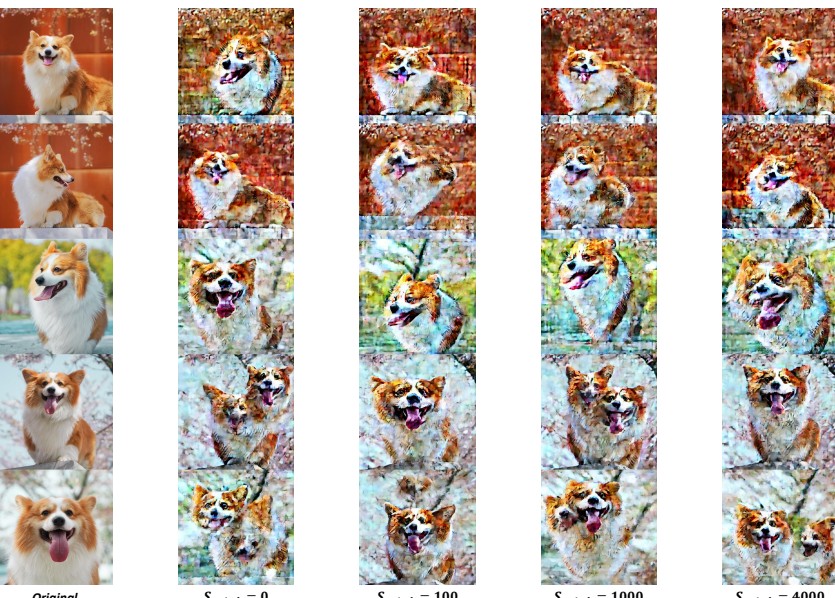

Figure 11: Visual comparison under different $S_{switch}$ settings. $S_{switch} = 0$ indicates that $L_{mix}$ is never activated, while larger values correspond to longer activation periods.

The results demonstrate a consistent pattern. When $L_{mix}$ is disabled entirely ($S_{switch} = 0$), reconstructed images display pronounced feature mixing and reduced structural coherence, indicating that $L_{mix}$ effectively suppresses cross-sample feature interference. If $L_{mix}$ remains active for the entire optimization (e.g., $S_{switch} = 4000$), feature disentanglement is stronger but some fine-grained details—such as textures and local contrast—tend to be muted, which can degrade certain metrics. Intermediate activation durations (e.g., $S_{switch} = 100$ or 1000) strike a favorable balance, preserving both feature separation and detail recovery and yielding more robust overall results.

In summary, $L_{mix}$ primarily serves as a regularizer that reduces feature mixing and improves visual separability between samples rather than uniformly boosting all quantitative metrics. Empirically, setting $S_{switch}$ to a moderate value offers a practical trade-off between feature disentanglement and detail preservation, leading to improved perceptual quality.

## K    RECOVERY UNDER DIFFERENT RANDOM SEEDS

To evaluate the robustness of our pipeline to different random initializations, we test both the first-stage method (GradCFG) and the complete two-stage pipeline (GradCFG + Inv-Sam) on the `backpack` category using multiple random seeds. For each seed we measure image reconstruction quality (SSIM / PSNR / LPIPS) and text-recovery consistency (embedding similarity and PSNR of images generated from recovered embeddings). Table 13 reports per-seed image reconstruction metrics for the two configurations, while Table 14 summarizes text-recovery stability.

Table 13: Image reconstruction under different seeds.

| Seed | GradCFG | | | + Inv-Sam | | |
|------|---------|---------|---------|-----------|---------|---------|
|      | SSIM ↑ | PSNR ↑ | LPIPS ↓ | SSIM ↑ | PSNR ↑ | LPIPS ↓ |
| seed 1 | 0.1933 | 11.780 | 0.7024 | 0.2862 | 12.770 | 0.5781 |
| seed 2 | 0.1895 | 11.230 | 0.7145 | 0.3149 | 12.190 | 0.5646 |
| seed 3 | 0.1834 | 11.390 | 0.7134 | 0.2732 | 12.000 | 0.6027 |
| **Avg.** | **0.1887** | **11.467** | **0.7101** | **0.2914** | **12.320** | **0.5818** |

Table 14: Stability of text recovery under different seeds (embedding similarity and generated-image PSNR).

| Seed | Similarity ↑ | PSNR ↑ |
|------|--------------|--------|
| seed 1 | 0.7555 | 16.73 |
| seed 2 | 0.7542 | 16.52 |
| seed 3 | 0.7556 | 16.87 |
| **Avg.** | **0.7551** | **16.707** |

The per-seed results indicate that GradCFG (stage 1) produces stable reconstructions across different random seeds: SSIM varies by about 0.0099 (0.1834–0.1933), PSNR by $\approx 0.55$ dB, and LPIPS by $\approx 0.0121$. The full two-stage method (GradCFG + Inv-Sam) yields higher average metrics but exhibits slightly larger sensitivity to the initialization: SSIM spans $\approx 0.0417$, PSNR $\approx 0.77$ dB, and LPIPS $\approx 0.0381$ across the three seeds. This behavior is consistent with the pipeline design—Inv-Sam refines the stage-1 result to improve visual fidelity, but the additional refinement steps increase dependence on initialization and optimization trajectory, leading to somewhat greater variance.

Text-recovery metrics are highly stable: the recovered prompt embeddings vary negligibly across seeds (embedding similarity differs by only $\approx 0.0014$), and the PSNR of images generated from those embeddings varies by about 0.35 dB. These observations suggest that the textual semantics recovered by our method are largely seed-insensitive and, when used to condition generation, produce consistent downstream images.

In summary, GradCFG provides a robust and repeatable stage-1 reconstruction across seeds, while the full two-stage pipeline consistently improves visual fidelity at the cost of slightly increased sensitivity to initialization.

## L    INV-SAM WITHOUT ACCESS TO THE FINE-TUNED MODEL

In practical scenarios, an attacker or researcher may not have access to the parameters of the fine-tuned target model $\theta_R$. To handle this more challenging setting, we further develop a variant of Inv-Sam that does not rely on the fine-tuned model. Instead, the method only uses the available model parameters $\theta_r$ (i.e., the base model before fine-tuning) to perform a post-hoc refinement step. The key idea is to leverage the generative capability of the base model itself to enhance the initially recovered image $x_0$ and text prompt $C_0$ produced by GradCFG, thereby improving visual fidelity and text–image consistency even in the absence of fine-tuned model priors.

Concretely, the procedure first performs a reverse diffusion process guided by an empty text prompt $C_\phi$ to remove artifacts and map the recovered image back into a more semantically stable latent region. It then performs a forward conditional sampling step guided by the recovered text $C_0$ to reinforce semantic content. Algorithm 3 summarizes the full procedure of Inv-Sam when the fine-tuned model is unavailable.

---

**Algorithm 3** Inv-Sam Optimization Using Only $\theta_r$

---

**Input:** Initial recovered latent $\hat{\mathbf{x}}_0 \in \mathbb{R}^m$; recovered text embedding $\hat{C}_0 \in \mathbb{R}^{77 \times 768}$
sampling-guidance scale $\omega_{\text{sam}}$
Noise schedule $\{\bar{\alpha}_t\}_{t=0}^T$
Available model parameters $\theta_r$

**Phase I: Reverse Diffusion (Artifact Removal)** $\quad \mathbf{x}_0^{\text{inv}} \leftarrow \hat{\mathbf{x}}_0$
**for** $t = 0$ **to** $T - 1$ **do**

$\quad \epsilon_{\text{empty}} \leftarrow \epsilon_{\theta_r}(\mathbf{x}_t^{\text{inv}}, t, C_\phi) \; \tilde{\epsilon}^{\text{inv}} \leftarrow \epsilon_{\text{empty}}$

$\quad \mathbf{x}_{t+1}^{\text{inv}} \leftarrow \sqrt{\bar{\alpha}_{t+1}} \left( \frac{\mathbf{x}_t^{\text{inv}} - \sqrt{1 - \bar{\alpha}_t} \, \tilde{\epsilon}^{\text{inv}}}{\sqrt{\bar{\alpha}_t}} \right) + \sqrt{1 - \bar{\alpha}_{t+1}} \, \tilde{\epsilon}^{\text{inv}}$

**Phase II: Conditional Sampling (Semantic Reinforcement)** $\quad \mathbf{x}_T^{\text{sam}} \leftarrow \mathbf{x}_T^{\text{inv}}$
**for** $t = T - 1$ **to** $0$ **do**

$\quad \epsilon_{\text{empty}} \leftarrow \epsilon_{\theta_r}(\mathbf{x}_{t+1}^{\text{sam}}, t + 1, C_\phi)$

$\quad \epsilon_{\text{text}} \leftarrow \epsilon_{\theta_r}(\mathbf{x}_{t+1}^{\text{sam}}, t + 1, \hat{C}_0)$

$\quad \tilde{\epsilon}^{\text{sam}} \leftarrow \epsilon_{\text{empty}} + \omega_{\text{sam}}(\epsilon_{\text{text}} - \epsilon_{\text{empty}})$

$\quad \mathbf{x}_t^{\text{sam}} \leftarrow \sqrt{\bar{\alpha}_t} \left( \frac{\mathbf{x}_{t+1}^{\text{sam}} - \sqrt{1 - \bar{\alpha}_{t+1}} \, \tilde{\epsilon}^{\text{sam}}}{\sqrt{\bar{\alpha}_{t+1}}} \right) + \sqrt{1 - \bar{\alpha}_t} \, \tilde{\epsilon}^{\text{sam}}$

**Output:** Final refined latent $\hat{\mathbf{x}}_0^{\text{opt}} \leftarrow \mathbf{x}_0^{\text{sam}}$

---

Figure 12 compares the enhancement effects of Inv-Sam under two settings: when the fine-tuned model is available, and when the fine-tuned model is unknown.

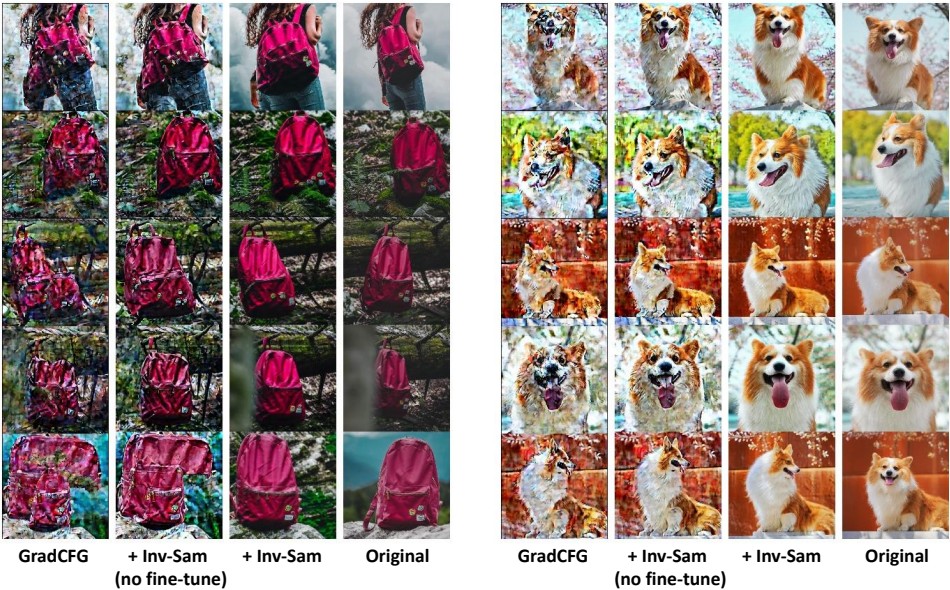

| GradCFG | + Inv-Sam (no fine-tune) | + Inv-Sam | Original | | GradCFG | + Inv-Sam (no fine-tune) | + Inv-Sam | Original |

Figure 12: Comparison of Inv-Sam with and without access to the fine-tuned model.

The results show that even without the fine-tuned model, Inv-Sam can still noticeably improve visual fidelity and local detail. Although the enhancement is less pronounced compared to using the fine-tuned model, this further confirms that the discrepancy between the fine-tuned and base models indeed provides additional useful priors for reconstructing the original training data.

## M   GRADCFG UNDER MULTI-CLASS FINE-TUNING

In this experiment, we evaluate the effectiveness of our method in a more challenging multi-class fine-tuning setting. Concretely, we fine-tune the diffusion model on several distinct object categories and then apply both GradCFG and Inv-Sam to reconstruct the underlying image and text data from gradients. For each method, we report three standard image similarity metrics (SSIM, PSNR, LPIPS) together with the cosine similarity between the reconstructed and original text embeddings.

Table 15 summarizes the quantitative reconstruction performance under this multi-category fine-tuning scenario, while Figure 13 presents visual examples of the recovered images. We observe that our approach can consistently recover meaningful images and texts across different object categories, and Inv-Sam further improves reconstruction quality over GradCFG alone. These results indicate that our gradient inversion attack is not restricted to single-category personalization, but remains effective and robust in more realistic multi-class fine-tuning scenarios.

Table 15: Image reconstruction performance of GradCFG and Inv-Sam under multi-class fine-tuning, evaluated at a given text embedding similarity.

| Text similarity | GradCFG | | | + Inv-Sam | | |
|---|---|---|---|---|---|---|
| | SSIM ↑ | PSNR ↑ | LPIPS ↓ | SSIM ↑ | PSNR ↑ | LPIPS ↓ |
| 0.7795 | 0.1364 | 10.460 | 0.7762 | 0.2033 | 11.520 | 0.6461 |

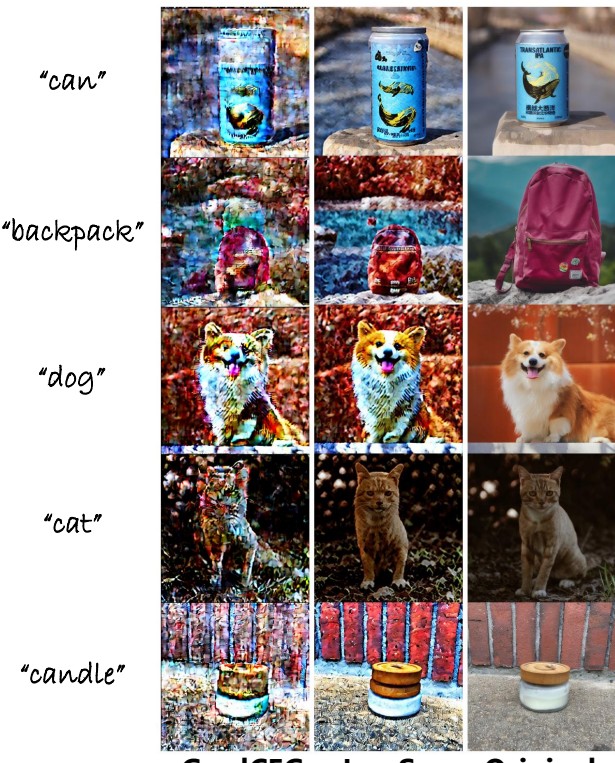

Figure 13: Qualitative results of gradient inversion attacks under multi-class fine-tuning.

