# OpenReview forum: "GradCFG: Gradient Inversion of Classifier-Free Guidance Diffusion Models"
_ICLR.cc/2026/Conference — Submitted to ICLR 2026_

### Official Review · Reviewer_aUCK · 2025-10-15

**Soundness:** 3
**Presentation:** 3
**Contribution:** 3
**Rating:** 4
**Confidence:** 3

**Summary:**

This paper investigates gradient inversion attacks on Classifier-Free Guided (CFG) diffusion models, aiming to expose potential privacy risks when model gradients are shared. The authors propose GradCFG, a framework that jointly optimizes four variables—the image latent, text embedding, noise, and diffusion timestep—to reconstruct both image and text data from gradients. The method introduces a continuous-time reparameterization to make the discrete timestep differentiable and a periodic noise reset to enhance optimization stability. Building upon this, the authors design Inv-Sam, an inverse–sampling refinement algorithm that leverages the difference between the fine-tuned and original models’ noise predictions to enhance reconstructed details. The approach effectively recovers private sample information and demonstrates a new privacy vulnerability in diffusion models, though the theoretical justification of Inv-Sam remains largely empirical.

**Strengths:**

1.	The paper tackles a novel and underexplored problem—gradient inversion attacks on Classifier-Free Guided (CFG) diffusion models—highlighting a significant and realistic privacy risk in generative model training.
2.	The follow-up Inv-Sam refinement strategy is conceptually creative, leveraging the difference between fine-tuned and base models to recover lost high-frequency details.
3.	Experiments on realistic datasets show clear improvements in reconstruction quality, and the analysis provides meaningful insights into multi-modal gradient inversion.

**Weaknesses:**

1.	The Inv-Sam refinement relies on an empirical assumption that the difference between the fine-tuned and base models encodes training-sample-specific details. While this idea is intuitively appealing, it lacks solid theoretical justification or causal evidence linking parameter shifts to concrete data features. The claim that Inv-Sam restores “fine-grained details” therefore appears somewhat overstated without stronger empirical validation. It is more plausible that the observed signal primarily reflects semantic adaptation—changes in concept alignment or attention—rather than genuine pixel-level detail recovery.
2.	The optimization process in GradCFG is extremely sensitive to the intrinsic stochasticity of diffusion models. Diffusion processes involve random noise sampling and discrete timesteps, both of which can drastically change the gradient landscape. Even though GradCFG introduces noise resetting and timestep reparameterization, the optimization objective remains non-convex, noisy, and highly multimodal. This means convergence may depend heavily on initialization and random seeds, making the reconstructed results unstable or inconsistent across runs.
3.	The computational cost of jointly optimizing four high-dimensional variables (latent image, text embedding, noise, and timestep) is considerable, limiting the approach’s practical scalability to larger CFG models or federated learning settings.

**Questions:**

Please see the weakness.

Given the inherently stochastic nature of diffusion processes—where random noise sampling and discrete timesteps play a central role—the reliability of any gradient-based inversion remains questionable. As a result, the reproducibility and stability of the proposed inversion process are uncertain, raising concerns about its robustness in practice.

---

> ### Author Response · Authors · 2025-11-22
> **Response to Reviewer aUCK (part1)**
>
> We appreciate your insightful questions regarding the interpretation of Inv-Sam, robustness to randomness, and the computational cost and scalability of our method. We respond as follows.
>
> ### R4.1 On the theoretical and causal interpretation of Inv-Sam
>
> Thank you for highlighting the issue about how Inv-Sam should be interpreted. We agree that Inv-Sam is more appropriately viewed as a mechanism for enhancing **semantic alignment** and **perceptual detail quality**, rather than as a method for exact pixel-level recovery of the training data.
>
> More concretely, Inv-Sam relies on an empirical assumption that the difference between the fine-tuned and base models encodes training-sample–specific signals. While this is intuitively appealing, we *do not claim a rigorous theoretical or causal link between specific parameter shifts and concrete data features*. In light of your comment, we now emphasize that the observed improvements are best understood as **semantic adaptation** and enhanced perceptual plausibility, rather than guaranteed recovery of “fine-grained” pixel-level details.
>
> In the revised manuscript, we have corrected and softened the relevant statements and removed phrases suggesting exact recovery of fine-grained details (including those around line 90 in the original submission).[1–3,6].
>
> ---
>
> ### R4.2 On sensitivity to randomness and robustness of the optimization
>
> Thank you for suggesting a systematic evaluation of robustness with respect to random seeds. In practice, we had observed that our method is relatively robust to initialization, which is why we did not initially emphasize this aspect. Following your comment, we conducted experiments with different random seeds and report the quantitative results below.
>
> For image reconstruction (GradCFG and Inv-Sam):
>
> *Table R8: Seed-wise image reconstruction metrics for GradCFG and Inv-Sam (higher SSIM/PSNR and lower LPIPS are better).*
>
> | Seed  | SSIM↑ (GradCFG) | PSNR↑ (GradCFG) | LPIPS↓ (GradCFG) | SSIM↑ (Inv-Sam) | PSNR↑ (Inv-Sam) | LPIPS↓ (Inv-Sam) |
> | ----- | --------------: | --------------: | ---------------: | --------------: | --------------: | ---------------: |
> | seed1 |          0.1933 |          11.780 |           0.7024 |          0.2862 |          12.770 |           0.5781 |
> | seed2 |          0.1895 |          11.230 |           0.7145 |          0.3149 |          12.190 |           0.5646 |
> | seed3 |          0.1834 |          11.390 |           0.7134 |          0.2732 |          12.000 |           0.6027 |
>
> For text-related reconstruction (embedding similarity and PSNR of generated images conditioned on recovered prompts):
>
> *Table R9: Seed-wise text-related reconstruction metrics (higher similarity and PSNR are better).*
>
> | Seed  | Similarity↑ | PSNR↑ |
> | ----- | -------------------- | -------------------------- |
> | seed1 | 0.7555               | 16.73                      |
> | seed2 | 0.7542               | 16.52                      |
> | seed3 | 0.7556               | 16.87                      |
>
> The variations across seeds are small, indicating that the method is not overly sensitive to random initialization. We summarize these results and additional discussion in Appendix K.

---

> ### Author Response · Authors · 2025-11-22
> **Response to Reviewer aUCK (part2)**
>
> ### R4.3 On computational cost and scalability
>
> To the best of our knowledge, our work is the first to systematically apply gradient inversion attacks to **CFG-based diffusion models** used in practical applications [4,5]. Before GradCFG, gradient attacks were primarily studied on classification networks and relatively small generative models [1,2,3,6]. In our setting, we must cope with:
>
> 1. Much **larger model architectures** (e.g., Stable Diffusion–style models) with extremely high-dimensional parameter spaces [4].
> 2. A more complex training paradigm with **random time steps**, **random noise**, and classifier-free guidance [4].
> 3. High-dimensional, multimodal recovery targets $(x, C, \epsilon, t)$ [4,6].
>
> These factors collectively make the optimization problem more challenging and computationally expensive. Under these conditions, the reconstruction quality we obtain is already the result of substantial algorithmic and engineering efforts.
>
> To provide evidence that our method is not limited to tiny models, we conducted few-shot experiments on **SD 1.4** [4] in the rebuttal (as also presented to Reviewers TnL3 and sgw7). We reproduce the results here for completeness:
>
> *Table R10: Image reconstruction metrics on SD 1.4 for GradCFG and Inv-Sam (higher SSIM/PSNR and lower LPIPS are better).*
>
> | label       | SSIM↑ (GradCFG) | PSNR↑ (GradCFG) | LPIPS↓ (GradCFG) | SSIM↑ (Inv-Sam) | PSNR↑ (Inv-Sam) | LPIPS↓ (Inv-Sam) |
> | ----------- | --------------: | --------------: | ---------------: | --------------: | --------------: | ---------------: |
> | backpack    |          0.1568 |          10.120 |           0.7645 |          0.4511 |          12.110 |           0.5011 |
> | cat         |          0.1219 |          10.860 |           0.6761 |          0.2367 |          11.590 |           0.5165 |
> | dog         |          0.2137 |          11.180 |           0.6995 |          0.2652 |          11.250 |           0.6071 |
> | monster toy |          0.1388 |          10.550 |           0.7541 |          0.3440 |          13.320 |           0.5120 |
> | robot toy   |          0.0821 |           9.120 |           0.7400 |          0.1415 |          10.370 |           0.5623 |
> | **Average** |      **0.1427** |     **10.3660** |        **0.7268** |      **0.2877** |      **11.7280** |        **0.5398** |
>
> *Table R11: Text reconstruction metrics on SD 1.4 (higher similarity and PSNR are better).*
>
> | label       | similarity↑ |  PSNR↑ |
> | ----------- | ---------: | ----: |
> | backpack    |     0.6538 | 13.47 |
> | cat         |     0.8174 | 15.60 |
> | dog         |     0.8641 | 17.20 |
> | monster toy |     0.6311 | 12.45 |
> | robot toy   |     0.6201 | 15.34 |
> | **Average** |   **0.7173** | **14.812** |
>
> These results show that both GradCFG and Inv-Sam can still produce meaningful reconstructions on the larger SD 1.4 model [4], indicating some degree of scalability. In future work, we plan to explore more efficient optimization strategies and architectural modifications to further reduce computational cost and improve the practical applicability of GradCFG.
>
>
> ---
>
> ### References
>
> [1] L. Zhu, Z. Liu, and S. Han. Deep Leakage from Gradients. NeurIPS, 2019.
>
> [2] J. Geiping, H. Bauermeister, H. Dröge, and M. Moeller. Inverting Gradients – How Easy Is It to Break Privacy in Federated Learning? NeurIPS, 2020.
>
> [3] H. Yin, A. Mallya, A. Vahdat, J. M. Alvarez, J. Kautz, and P. Molchanov. See Through Gradients: Image Batch Recovery via GradInversion. CVPR, 2021.
>
> [4] R. Rombach, A. Blattmann, D. Lorenz, P. Esser, and B. Ommer. High-Resolution Image Synthesis with Latent Diffusion Models. CVPR, 2022.
>
> [5] N. Ruiz, Y. Li, V. Jampani, Y. Pritch, M. Rubinstein, and K. Aberman. DreamBooth: Fine Tuning Text-to-Image Diffusion Models for Subject-Driven Generation. CVPR, 2023.
>
> [6] J. Huang, C. Hong, S. Roos, and L. Y. Chen. GIDM: Gradient Inversion of Federated Diffusion Models. ARES, 2025.

---

> ### Comment · Reviewer_aUCK · 2025-11-26
> **response to authors**
>
> Thanks for the rebuttal and experiment. However, I still think the random generation process is not fully tested, e.g., more timesteps settings.

---

> ### Author Response · Authors · 2025-11-27
> **Response to Reviewer aUCK**
>
> We appreciate your positive feedback on our other responses and additional experiments, and we thank you for raising this important question. As you correctly pointed out, CFG-based gradient inversion attacks indeed exhibit stochasticity, and the choice of timesteps is a critical factor. Regarding your comment on the timestep setting, our understanding of this notion can be roughly divided into two main aspects: (i) the configuration of the diffusion timestep scheduler itself, and (ii) the setting of the sampled timesteps during fine-tuning. These two aspects constitute the most central components of the overall timestep setting, and we address them in turn below.
>
>
> ### (1) Timestep scheduler configuration
>
> Regarding the timestep scheduler aspect, the timestep setting includes the configuration of the diffusion scheduler, such as the total number of diffusion steps and the choice of the $\beta$-schedule. While different scheduler designs may in principle influence the behavior of gradient inversion, in our fine-tuning setup the scheduler is required to remain identical to that of the pretrained diffusion model. In practice, this means that all scheduler hyperparameters are fixed by the pretrained model and are not modified during fine-tuning.
>
> Importantly, the timestep reparameterization strategy proposed in GradCFG is not tailored to a single, specific scheduler configuration; rather, it is a generic mechanism that can, in principle, be combined with different diffusion schedulers. Although for fairness and consistency all our experiments use the same scheduler as the pretrained model, the method itself is designed to be broadly applicable beyond this particular setting.
>
> ### (2) Effect of sampled training timesteps on reconstruction quality
>
> For the timestep setting in terms of training timesteps sampled during fine-tuning, we have also conducted a dedicated analysis. Concretely, we partition the training timesteps (i.e., the original training timesteps $t_0$ used during fine-tuning) into three ranges:
>
> - early timesteps: $(0, 200)$
> - middle timesteps: $(200, 800)$
> - late timesteps: $(800, 1000)$
>
> For each range, we sample timesteps during fine-tuning, obtain the corresponding gradients, and then perform gradient inversion attacks. The reconstruction performance of GradCFG and Inv-Sam under these different timestep ranges is summarized in **Table R.12**.
>
> **Table R.12.** Reconstruction quality under different training timestep ranges for GradCFG and Inv-Sam.
>
> | timesteps range | SSIM↑ (GradCFG) | PSNR↑ (GradCFG) | LPIPS↓ (GradCFG) | SSIM↑ (Inv-Sam) | PSNR↑ (Inv-Sam) | LPIPS↓ (Inv-Sam) |
> | ----------------------- | --------------: | --------------: | ---------------: | --------------: | --------------: | ---------------: |
> | 0–200                   |          0.3240 |          10.95  |           0.6164 |          0.4505 |          12.85  |           0.4997 |
> | 200–800                 |          0.1997 |          10.70  |           0.7298 |          0.3149 |          12.56  |           0.5593 |
> | 800–1000                |          0.1603 |           9.07  |           0.8198 |          0.1567 |           9.84  |           0.8006 |
>
> From Table R.12, we observe that GradCFG achieves relatively good reconstruction quality when the training timesteps lie in the range $t \in [0, 800]$. In this region, the method is reasonably robust to the choice of timestep range. The performance degrades more noticeably when timesteps lie in the late range $t \in [800, 1000]$.
>
> This degradation is consistent with the underlying diffusion process. When $t$ is in the late regime, the corresponding scheduling term $\sqrt{\bar{\alpha}_t}$ becomes very close to zero. In CFG training objective, the clean image $x_0$ is only provided to the noise predictor through the noised combination $\sqrt{\bar{\alpha}_t} x_0 + \sqrt{1 - \bar{\alpha}_t}\epsilon$. As $\sqrt{\bar{\alpha}_t} \to 0$, this input becomes almost entirely dominated by the Gaussian noise $\epsilon$ and contains almost no information about $x_0$. As a consequence, the gradients themselves carry much weaker signals about the underlying image, making gradient inversion inherently difficult—if not impossible—for very large $t$. This limitation stems from the diffusion dynamics and therefore affects *any* gradient inversion method, not just ours.
>
>
> On the other hand, in the more practically relevant range $t \in [0, 800]$, GradCFG shows stable reconstruction performance across different timestep ranges, indicating that our method is reasonably robust with respect to timestep sampling in typical training settings.
>
> Once again, we thank you for highlighting this subtle but important point. If we have misunderstood any aspect of your comment regarding the timestep setting, or if you have further questions or suggestions, we would be very happy to continue the discussion.

---

### Official Review · Reviewer_xW4i · 2025-10-29

**Soundness:** 3
**Presentation:** 2
**Contribution:** 2
**Rating:** 4
**Confidence:** 4

**Summary:**

This paper adapts the gradient inversion attack, which is commonly used in classifier models, to the domain of diffusion generation models. To demonstrate the feasibility of this attack on Classifier-Free Guidance (CFG) models, the authors propose GradCFG, complemented by the Inv-Sam strategy. GradCFG is designed around a Quadruple Collaborative Optimization Algorithm, which enables the reconstruction of multiple high-dimensional variables: the image latent variable $x_0$, text embeddings $C_0$, noise $\epsilon$, and reparameterized continuous time steps $t$. The Inv-Sam strategy further enhances the quality of the reconstructed image latent variable by leveraging the differences between the pre-finetuning and post-finetuning versions of the model.

**Strengths:**

1. The evaluation includes detailed experiments that sufficiently demonstrate the effectiveness and efficacy of the proposed GradCFG method.
2. The proposed method successfully recovers both the training image data and the paired text prompt simultaneously, addressing a significant challenge in prior gradient inversion attacks.
3. The implementation details of the method are clearly explained and effectively supported by corresponding pseudocode, contributing to the reproducibility of the work.

**Weaknesses:**

1. The paper claims to present a gradient inversion attack specifically applicable to general Classifier-Free Guidance (CFG) diffusion models. However, the experimental validation is exclusively performed on personalized generation tasks. This limited scope suggests the claim might be overly broad.
2. The clarity and focus of the writing in certain sections are weak (e.g., the content described around Line 80 is overly broad and lacks necessary specificity).
3. The paper fundamentally emphasizes a gradient attack methodology. However, the Inv-Sam module utilizes the output and knowledge derived from the finetuned model. The authors should provide a robust justification for the consistency and validity of incorporating this model-specific information within the definition of a gradient inversion attack.
4. The experiment presented in Section 5.4 intended to demonstrate the non-uniqueness of the noise $\epsilon$ is not convincing. The large discrepancy observed between $\hat{\epsilon}$ and $\epsilon$ could potentially be attributed to the model's inherent prediction accuracy limitations or the intrinsic stochasticity of $\epsilon$ itself, rather than solely being definitive proof of its non-uniqueness.

**Questions:**

1. Since the Inv-Sam module relies on priors from the fine-tuned model, it is important to clarify how much the model overfits to the training data. Please provide visualizations of the fine-tuned model to support this analysis.
2. The current experimental setting focuses on attacking models fine-tuned using a single image. In practical personalization tasks, multiple images are typically used for fine-tuning. Would the proposed GradCFG method remain effective under these more complex multi-image fine-tuning conditions?
3. The ablation study is incomplete. Specifically, ablation studies are missing for key hyperparameters within the optimization process, such as the switching time step ($S_{\text{switch}}$) and the reset time step ($S_{\text{reset}}$).
4. The function of the $L_{\text{mix}}$ loss appears limited to suppressing uncorrelated noise in the reconstructed $x_0$, rather than achieving the stated goal of “feature disentanglement and prevention of feature mixing.” The authors should provide further ablation studies or visual analyses to substantiate the claimed disentanglement effect.
5. It is mentioned in the supplementary materials that the continuous time step $t$ is restricted to the range [400, 600] during optimization. What is the fundamental justification of this specific constraint?

---

> ### Author Response · Authors · 2025-11-22
> **Response to Reviewer xW4i (part1)**
>
> Thank you for your detailed questions regarding the scope of claims, clarity of writing, and the experimental justification of our design choices. We respond to each point below.
>
> ### R3.1 On whether the claimed scope is too broad
>
> GradCFG is **not** inherently restricted to fine-tuning tasks. It is designed around a **general CFG training paradigm** [4] and, in principle, can be applied to other CFG-based training scenarios beyond personalization. At the same time, we clearly stated in both the abstract and the introduction that the empirical study in this paper is conducted in **fine-tuning–based personalized generation** settings.
>
> Importantly, to our knowledge, our work is the first to extend gradient inversion attacks to large-scale CFG models. Before GradCFG, most gradient inversion attacks primarily targeted classification models or relatively small generative models [1–3,6]. Our setting:
> - Have more complex modalities (image + text).
> - Use more complex training with random time steps and noise.
> - Have substantially larger parameter counts [4].
>
> These factors significantly increase both the difficulty and the computational cost of reconstruction. Given these challenges and the fact that ours is, to the best of our knowledge, the first such attack on CFG models, we chose a DreamBooth-like **few-shot personalization** setup [5] as our primary experimental scenario to strike a balance between realism and feasibility under limited computational resources.
>
> Furthermore, this setup is not a toy example: DreamBooth-style personalization is widely used in practice and thus represents a realistic and practically relevant use case. At the same time, we agree that our current experiments cover a limited range of tasks. In the revised manuscript, we further sharpen the wording in the abstract and introduction to more precisely state that our empirical study focuses on fine-tuning–based personalization, and we explicitly highlight extending GradCFG to broader CFG application scenarios as an important direction for future work.
>
> ---
>
> ### R3.2 On evidence and explanation for the non-uniqueness of noise $\epsilon$
>
> Thank you for pointing to our discussion of noise non-uniqueness in Section 5.4. The experiments in that section are intended to provide **statistical evidence** that exact reconstruction of $\epsilon$ is not necessary for successful gradient alignment.
>
> Empirically, across many trials, we observe that even when the similarity between the recovered noise $\hat{\epsilon}$ and the ground-truth noise $\epsilon$ in pixel space is relatively low (e.g., below 0.3), the corresponding gradient vectors induced by these two noises are still highly aligned, with gradient similarity (measured as cosine similarity between flattened gradient vectors) often exceeding 95%. This suggests that there exists a large “feasible set” of $\epsilon$ values that yield nearly identical gradients. This observation supports the practical effectiveness of our periodic noise reset strategy: we do not need to chase the exact ground-truth $\epsilon$, as long as we can maintain gradient-level agreement.
>
> Importantly, this phenomenon is not specific to Tiny-SD; we observe similar behavior on larger models such as SD 1.4 [4], which indicates that it is likely a more general property rather than an artifact of a particular architecture.
>
> We also acknowledge that our current understanding is primarily empirical and does not yet provide a complete theoretical explanation. In the revised manuscript, we will make it clearer that the purpose of this analysis is to justify the practical effectiveness and reasonableness of the noise reset strategy, while a deeper theoretical study of $\epsilon$ non-uniqueness is an interesting open problem for future work.
>
>
> ---
>
> ### R3.3 On clarity of writing
>
> Thank you for pointing out that some paragraphs are dense and not sufficiently clear. We have rewritten and streamlined several key paragraphs, including those around line 80 in the original submission, which discuss our motivation and challenges.
>
> In the revised version, we focus these paragraphs on two central points:
>
> 1. Why gradient inversion on CFG models is fundamentally more challenging than in previous settings (larger models, multimodal outputs, random time steps and noise) [1–4,6].
> 2. How GradCFG is designed to tackle the core difficulties of multi-sample, multimodal, and high-dimensional recovery.
>
> These changes have been applied in the revision.
>
> ---

---

> ### Author Response · Authors · 2025-11-22
> **Response to Reviewer xW4i (part2)**
>
> ### R3.4 On the use of the fine-tuned model in Inv-Sam and overfitting concerns
>
> This is a crucial point, and we appreciate you raising it.
>
> We fully agree that using a **fine-tuned model** introduces additional prior information that does not come purely from gradients. Our intention is not to claim that Inv-Sam alone can “recover the exact training data” from gradients. Rather, Inv-Sam is explicitly designed as an **auxiliary module** that improves the **visual quality** and **semantic alignment** of the images already reconstructed by GradCFG.
>
> #### (1) **GradCFG** (first stage, not rely on fine-tuned models) plays most important role.
>
> From a technical perspective, the core difficulty and main innovation of our work lie in the **first stage**, i.e., GradCFG:
>
> - It must cope with the multi-solution nature induced by random noise and discrete time steps in diffusion training;
> - It must disentangle the contributions of different samples within aggregated gradients;
> - It targets high-dimensional, multimodal recovery (images, text, noise, and time steps).
>
> In this stage, GradCFG uses only the gradients and the current model parameters; it does **not** rely on any fine-tuned model. Without GradCFG’s initial recovery of $x_0$ and $C_0$, Inv-Sam cannot be applied at all.
>
> #### (2) **Inv-Sam** (second stage) variants without access to fine-tuned models works.
>
> To address the realistic scenario where an attacker may **not** have access to the fine-tuned model $\theta_R$, we additionally designed and implemented a variant that uses only the **current model** $\theta_r$:
>
> - After GradCFG produces initial estimates $x_0$ and $C_0$, we first run a reverse diffusion process guided by an **empty prompt** $C_{\phi}$;
> - We then run a forward diffusion process guided by the recovered text $C_0$ to obtain a **semantically enhanced** image.
>
> This refinement uses only the generative capability of the available model $\theta_r$ and does not assume access to $\theta_R$. We describe this variant in detail in Appendix L and provide qualitative visual examples. The results show that, even without a fine-tuned model, this refinement can still noticeably improve visual quality, though the effect is weaker than when using $\theta_R$. This performance gap itself highlights that fine-tuning indeed strengthens the model’s ability to generate the specific concept [5].
>
> #### (3) Relation to prior work
>
> Moreover, using fine-tuned models or stronger external priors to assist recovery is a **common setting** in related literature. For example:
>
> - **Generative model prior** methods use pretrained generative models to regularize the initial image distribution [7].
> - **GIDM** generates an image from the same class with a trained model as an initial point for recovery [6].
>
> In federated learning scenarios, an adversarial server often has access to client-updated (fine-tuned) models or at least a good approximation of the data distribution [1,2,6]. Therefore, treating a fine-tuned model as a prior for Inv-Sam is consistent with realistic threat models considered in prior work, rather than an overly optimistic assumption.
>
> ---
>
> ### R3.5 On multi-image fine-tuning scenarios
>
> We clarify that our experiments are **not** restricted to single-image fine-tuning. Several results in Section 5 are based on multi-image fine-tuning scenarios with a batch size of 5, which is closer to practical DreamBooth-like settings [5]. We will explicitly clarify this in the main text to avoid giving the impression that all experiments are single-image experiments.

---

> ### Author Response · Authors · 2025-11-22
> **Response to Reviewer xW4i (part3)**
>
> ### R3.6 On ablations and the role of $L_{\text{mix}}$
>
> In Section 5.3, we have already provided qualitative ablations on the reset step $S_{\text{rest}}$. The loss term $L_{\text{mix}}$ is introduced to reduce feature mixing across samples by minimizing the **average pairwise similarity** among reconstructed images. Its goal is not merely to suppress noise, but specifically to disentangle features of different samples.
>
> In practice, $L_{\text{mix}}$ plays a **supporting** rather than **central** role: it refines the results but is not essential to making the attack work.
>
> To better illustrate its effect, we report an ablation on $S_{\text{switch}}$, which controls when $L_{\text{mix}}$ is turned on:
>
> *Table R7: Ablation on $S_{\text{switch}}$ controlling when $L_{\text{mix}}$ is enabled (higher SSIM/PSNR and lower LPIPS are better).*
>
> | $S_{\text{switch}}$ |  SSIM↑ |  PSNR↑ | LPIPS↓ |
> | -------------------: | -----: | -----: | -----: |
> |                    0 | 0.2486 | 11.210 | 0.7021 |
> |                  100 | 0.2655 | 11.280 | 0.6877 |
> |                 1000 | 0.2639 | 11.380 | 0.6940 |
> |                 4000 | 0.2522 | 11.170 | 0.6828 |
>
> These results show that enabling $L_{\text{mix}}$ for a reasonable duration (e.g., $S_{\text{switch}} = 100$ or 1000) can mitigate feature mixing and yield moderate performance gains, while enabling it for too long (e.g., 4000) may slightly harm fine-detail reconstruction. We provide additional visualizations and analysis in Appendix J.
>
>
> ---
>
> ### R3.7 On restricting the optimized time steps $t$
>
> Empirically, we observe that when the time step $t$ lies in the interval $[400, 600]$, the corresponding diffusion coefficient $\alpha_t$ (which reflects the noise level) covers most of the effective range between 0 and 1 in our noise schedule [4]. This gives a rich set of candidate states while keeping the search space manageable.
>
> Restricting $t$ to this interval thus strikes a practical balance between:
>
> - Having a sufficiently diverse set of candidate states for optimization,
> - Maintaining optimization stability and efficiency.
>
> This interval is **not** a strict theoretical requirement but an empirically chosen range that is reasonably robust in our setting. For different models or noise schedules, the optimal range may shift slightly.
>
> ---
>
> ### References
>
> [1] L. Zhu, Z. Liu, and S. Han. Deep Leakage from Gradients. NeurIPS, 2019.
>
> [2] J. Geiping, H. Bauermeister, H. Dröge, and M. Moeller. Inverting Gradients – How Easy Is It to Break Privacy in Federated Learning? NeurIPS, 2020.
>
> [3] H. Yin, A. Mallya, A. Vahdat, J. M. Alvarez, J. Kautz, and P. Molchanov. See Through Gradients: Image Batch Recovery via GradInversion. CVPR, 2021.
>
> [4] R. Rombach, A. Blattmann, D. Lorenz, P. Esser, and B. Ommer. High-Resolution Image Synthesis with Latent Diffusion Models. CVPR, 2022.
>
> [5] N. Ruiz, Y. Li, V. Jampani, Y. Pritch, M. Rubinstein, and K. Aberman. DreamBooth: Fine Tuning Text-to-Image Diffusion Models for Subject-Driven Generation. CVPR, 2023.
>
> [6] J. Huang, C. Hong, S. Roos, and L. Y. Chen. GIDM: Gradient Inversion of Federated Diffusion Models. ARES, 2025.
>
> [7] Jeon J, Lee K, Oh S, et al. Gradient inversion with generative image prior. NeurIPS, 2021.

---

> > ### Comment · Reviewer_xW4i · 2025-11-27
> >
> > Thank you for the response. I still have two questions for further clarification:
> >
> > 1.I agree that gradient-based attacks on diffusion models are more challenging. However, using experiments on DreamBooth-like personalization to claim that the proposed method can generalize to all diffusion training tasks (e.g., pre-training, fine-tuning) is not well-supported. For example, the data assumptions are different: personalization requires multiple images of the same subject, while image editing and image composition do not rely on such subject-consistent training data.
> >
> > 2.More details about Figure 6 are necessary. How many samples were used to compute the statistics? Also, since the authors claim results on SD 1.4, those should be included. Moreover, the revision statement — “we will make it clearer that the purpose of this analysis is to justify the practical effectiveness and reasonableness of the noise reset strategy” — does not seem to appear in the manuscript. Where exactly is this explained?

---

> > > ### Author Response · Authors · 2025-11-28
> > > **Response to Reviewer xW4i (part 1)**
> > >
> > > We sincerely thank you for your careful reading of our paper, your positive assessment of our experiments and explanations, and your thoughtful suggestions for further improvement. Below, we address your main concerns in detail and describe the additional experiments and clarifications we have incorporated into the revised version.
> > >
> > > ---
> > >
> > > ### 1. On the scalability of our method
> > >
> > > We appreciate your recognition of the difficulty of our task, and we fully understand your concern regarding the scalability of our method: since CFG-based models are used in a wide range of training scenarios, evaluating only on object-centric fine-tuning may seem insufficient to demonstrate general applicability.
> > >
> > > From a technical perspective, GradCFG is derived from the generic training paradigm of CFG models. Once access to training gradients is available, our attack can in principle be constructed to recover data, and it is not inherently restricted to single-category object fine-tuning. Following your suggestion, we further designed an experiment in a multi-class fine-tuning setting to more directly validate the effectiveness of our method when the fine-tuned categories differ.
> > >
> > > Concretely, we select one image from each of 5 different object categories as fine-tuning data. On top of this multi-class fine-tuning, we apply both GradCFG and Inv-Sam to reconstruct the underlying images and texts. For each method, we report:
> > > - the average cosine similarity between reconstructed and ground-truth text embeddings, and
> > > - three standard image similarity metrics: SSIM, PSNR, and LPIPS.
> > >
> > > The results are summarized in *Table R13* below.
> > >
> > > *Table R13: Image and text reconstruction performance of GradCFG and Inv-Sam under multi-class fine-tuning.*
> > >
> > > | Text similarity | SSIM↑ (GradCFG) | PSNR↑ (GradCFG) | LPIPS↓ (GradCFG) | SSIM↑ (Inv-Sam) | PSNR↑ (Inv-Sam) | LPIPS↓ (Inv-Sam) |
> > > | --------------- | --------------: | --------------: | ---------------: | --------------: | --------------: | ---------------: |
> > > | 0.7795          |          0.1364 |          10.460 |           0.7762 |          0.2033 |          11.520 |           0.6461 |
> > >
> > > From Table R13, we observe that even in this multi-class fine-tuning scenario, both GradCFG and Inv-Sam are able to produce reasonable image reconstructions, and the text embeddings can be recovered with relatively high cosine similarity (≈ 0.78). This indicates that our gradient inversion attack remains effective beyond single-category personalization, posing a privacy risk even when the fine-tuning data spans diverse object categories. Additional analyses and qualitative visualizations for this setting are provided in Appendix M.
> > >
> > > More broadly, since GradCFG is built on the general training framework of CFG models, its success in fine-tuning settings suggests that other CFG-based training regimes may also, in principle, be vulnerable to similar privacy risks. At the same time, we fully acknowledge that we cannot claim universal effectiveness without further empirical validation. As you correctly point out, attacking pre-training or large-scale fine-tuning with substantially more data is significantly more challenging, to the extent that even prior gradient inversion attacks on classification models struggle to fully handle the large-batch-size regime.
> > >
> > > Overall, recovering training data from gradients in high-dimensional, non-convex diffusion training is already highly non-trivial. Our current work provides a concrete and systematic demonstration of such risks in both personalized and multi-class fine-tuning scenarios, and we view it as an important first step toward understanding privacy leakage in more general CFG-based training pipelines.

---

> > > ### Author Response · Authors · 2025-11-28
> > > **Response to Reviewer xW4i (part 2)**
> > >
> > > ### 2. Additional clarification on the non-uniqueness of $\epsilon$
> > >
> > > Thank you very much for pointing out this issue. We agree that the original manuscript did not clearly explain this part, and we have revised the main paper accordingly. In particular, we have updated Section 5.4 of the paper as follows:
> > >
> > > 1. Clarifying the experimental setup and adding SD 1.4 results.
> > >    We now explicitly describe the experimental protocol used to study the non-uniqueness of $\epsilon$, and we additionally report results on the SD 1.4 model. Specifically, for each model we sample 100 different random initializations of $\epsilon$ and optimize them under fixed $(\mathbf{x}_0, C_0, t)$. During optimization, we record:
> > >    - the similarity between the recovered noise $\hat{\epsilon}$ and the ground-truth noise $\epsilon$, and
> > >    - the similarity between the simulated gradient $g$ and the original gradient $g_0$.
> > >    We then compute the mean and standard deviation of these similarities across all runs and visualize their evolution. The results show that, even when the gradients are highly aligned (e.g., similarity close to 1), the similarity between $\hat{\epsilon}$ and $\epsilon$ remains relatively low. This empirically supports the claim that $\epsilon$ is non-unique: multiple distinct noise configurations can yield nearly identical gradients.
> > >
> > > 2. Explaining the connection between $\epsilon$ non-uniqueness and the periodic reset strategy.
> > >    We further clarify why this non-uniqueness motivates our periodic reinitialization of $\epsilon$. Since different $\epsilon$ values can lead to similar gradients, the remaining variables $(\mathbf{x}_0, C_0, t)$ must be optimized such that gradient alignment holds robustly across multiple feasible $\epsilon$ configurations, rather than overfitting to a single noise realization. To encourage this robustness, our reconstruction procedure periodically resets $\epsilon$ and re-optimizes it during training. This forces $(\mathbf{x}_0, C_0, t)$ to repeatedly adapt to different plausible noise samples, which in practice leads to more stable and accurate reconstructions.
> > >
> > > ---
> > >
> > > Once again, we thank you for the insightful comments and constructive suggestions, and we welcome any further discussion or feedback.

---

### Official Review · Reviewer_sgw7 · 2025-10-30

**Soundness:** 2
**Presentation:** 2
**Contribution:** 2
**Rating:** 4
**Confidence:** 2

**Summary:**

This paper proposes GradCFG and Inv-Sam to address gradient inversion in classifier-free guidance diffusion models, aiming for joint image-text reconstruction. Experiments on DREAMBOOTH-like fine-tuning evaluate performance under generic/specific prompts, using metrics like SSIM and cosine similarity.

**Strengths:**

It is the first work to empirically demonstrate joint image-text gradient inversion for CFG models, filling a gap in diffusion model privacy research.

**Weaknesses:**

1. The experiments lack direct comparisons with existing diffusion model gradient inversion methods (e.g., Huang et al. 2025a) or state-of-the-art gradient inversion techniques (e.g., GradInversion), making it hard to gauge relative performance.
2. Only validates on TinySD and 512×512 DREAMBOOTH data; no tests on other CFG models (e.g., full Stable Diffusion) or higher resolutions.

**Questions:**

1. Why not compare with Huang et al. (2025a, GIDM) or other diffusion-based gradient inversion methods to highlight GradCFG/Inv-Sam’s advantages?
2. Is there a systematic rationale for choosing the Inv-Sam guidance scale ω_sam (e.g., hyperparameter sensitivity analysis)?

---

> ### Author Response · Authors · 2025-11-22
> **Response to Reviewer sgw7 (part1)**
>
> We greatly appreciate your detailed comments regarding comparisons with existing methods, generalization, and hyperparameter sensitivity. We respond point by point below.
>
> ---
>
> ### R2.1 Comparison with existing diffusion / gradient inversion methods
>
> Thank you for pointing out two related gradient inversion attack methods [3,6]. Broadly speaking, existing works mainly target classification networks or small diffusion models [1,2,3,6], whereas our work considers larger-scale **CFG-based** diffusion models and the joint recovery of **images and text prompts**.
>
> More concretely:
>
> - **GIDM** is designed for gradient inversion on a small **unconditional** diffusion model. The recovered data do not include text prompts, which makes it difficult to serve as a direct baseline for our multimodal CFG setting [6].
> - **GradInversion** is a method for gradient inversion on **classification networks**, whose supervised training paradigm (cross-entropy classification) is fundamentally different from CFG-based diffusion training [3].
>
> As a result, there is no existing method that can be used as a drop-in baseline in our exact setting. To nevertheless provide a comparable reference on image recovery, we adapted GIDM by explicitly providing it with the ground-truth prompt as prior information. We then compared its reconstruction quality against GradCFG, which is attacked **without access to the prompt**:
>
> *Table R4: Image reconstruction comparison between the adapted GIDM baseline and GradCFG (higher SSIM/PSNR and lower LPIPS are better).*
>
> | Method                          |  SSIM↑ | PSNR↑ | LPIPS↓ |
> | ------------------------------- | -----: | ----: | -----: |
> | GIDM (with ground-truth prompt) | 0.0564 | 8.273 | 0.8566 |
> | GradCFG (without prompt)        | 0.1240 | 10.60 | 0.7778 |
>
> The results show that GradCFG clearly outperforms the adapted GIDM baseline despite operating under a less favorable information setting. The full setup, additional metrics, and visualizations are provided in Appendix I. Taken together, these results show that the effectiveness of GradCFG in this challenging CFG setting is far from trivial, as it surpasses a strong baseline that is given strictly more information.
>
> ---
>
> ### R2.2 On generalization to larger models
>
> Your suggestion to explicitly examine generalization is very valuable. Conceptually, extending to larger models or higher resolutions primarily enlarges the **size and dimensionality** of the optimization problem. Within limited time and computational resources, we chose to focus our rebuttal experiments on **scaling up the model size**.
>
> As described in our response to Reviewer TnL3 , we conducted joint image-and-text recovery experiments on **SD 1.4** [4] using 2 images per prompt. The results are:
>
> *Table R5: Image reconstruction metrics on SD 1.4 for GradCFG and Inv-Sam (higher SSIM/PSNR and lower LPIPS are better).*
>
> | label       | SSIM↑ (GradCFG) | PSNR↑ (GradCFG) | LPIPS↓ (GradCFG) | SSIM↑ (Inv-Sam) | PSNR↑ (Inv-Sam) | LPIPS↓ (Inv-Sam) |
> | ----------- | --------------: | --------------: | ---------------: | --------------: | --------------: | ---------------: |
> | backpack    |          0.1568 |          10.120 |           0.7645 |          0.4511 |          12.110 |           0.5011 |
> | cat         |          0.1219 |          10.860 |           0.6761 |          0.2367 |          11.590 |           0.5165 |
> | dog         |          0.2137 |          11.180 |           0.6995 |          0.2652 |          11.250 |           0.6071 |
> | monster toy |          0.1388 |          10.550 |           0.7541 |          0.3440 |          13.320 |           0.5120 |
> | robot toy   |          0.0821 |           9.120 |           0.7400 |          0.1415 |          10.370 |           0.5623 |
> | **Average** |      **0.1427** |     **10.3660** |        **0.7268** |      **0.2877** |      **11.7280** |        **0.5398** |
>
> *Table R6: Text reconstruction metrics on SD 1.4 (higher similarity and PSNR are better).*
>
> | label       | similarity↑ |  PSNR↑ |
> | ----------- | ---------: | ----: |
> | backpack    |     0.6538 | 13.47 |
> | cat         |     0.8174 | 15.60 |
> | dog         |     0.8641 | 17.20 |
> | monster toy |     0.6311 | 12.45 |
> | robot toy   |     0.6201 | 15.34 |
> | **Average** |   **0.7173** | **14.812** |
>
> These results indicate that our method maintains meaningful performance when scaled to the larger SD 1.4 model [4], supporting the generalization of GradCFG with respect to model size. Detailed settings and visualizations are included in Appendix H.
>
> ---

---

> ### Author Response · Authors · 2025-11-22
> **Response to Reviewer sgw7 (part2)**
>
> ### R2.3 On the choice and sensitivity of the Inv-Sam guidance scale $\omega_{\text{sam}}$
>
> Our original submission already includes a sensitivity analysis of the guidance scale $\omega_{\text{sam}}$ in Appendix G, and we summarize the main observations here.
>
> Intuitively, $\omega_{\text{sam}}$ controls how much information is transferred from the fine-tuned model to the generation process:
>
> - When $\omega_{\text{sam}} = 0$, Inv-Sam **degenerates** to simply returning the image recovered by GradCFG, without any further refinement.
> - As $\omega_{\text{sam}}$ increases, the generated images become more influenced by the fine-tuned model, which typically improves semantic alignment and style consistency.
> - If $\omega_{\text{sam}}$ is too large, the refinement may overfit to the fine-tuned model, potentially drifting away from the gradient-derived reconstruction.
>
> Within a reasonable range, we observe that the method is relatively robust to changes in $\omega_{\text{sam}}$, and the performance does not fluctuate dramatically.
>
>
> ---
>
> ### Reference
>
> [1] L. Zhu, Z. Liu, and S. Han. Deep Leakage from Gradients. NeurIPS, 2019.
>
> [2] J. Geiping, H. Bauermeister, H. Dröge, and M. Moeller. Inverting Gradients – How Easy Is It to Break Privacy in Federated Learning? NeurIPS, 2020.
>
> [3] H. Yin, A. Mallya, A. Vahdat, J. M. Alvarez, J. Kautz, and P. Molchanov. See Through Gradients: Image Batch Recovery via GradInversion. CVPR, 2021.
>
> [4] R. Rombach, A. Blattmann, D. Lorenz, P. Esser, and B. Ommer. High-Resolution Image Synthesis with Latent Diffusion Models. CVPR, 2022.
>
> [5] N. Ruiz, Y. Li, V. Jampani, Y. Pritch, M. Rubinstein, and K. Aberman. DreamBooth: Fine Tuning Text-to-Image Diffusion Models for Subject-Driven Generation. CVPR, 2023.
>
> [6] J. Huang, C. Hong, S. Roos, and L. Y. Chen. GIDM: Gradient Inversion of Federated Diffusion Models. ARES, 2025.

---

### Official Review · Reviewer_TnL3 · 2025-11-01

**Soundness:** 3
**Presentation:** 4
**Contribution:** 3
**Rating:** 6
**Confidence:** 4

**Summary:**

This work presents the first gradient inversion attack targeting Classifier-Free Guidance (CFG) diffusion models. Unlike traditional settings that only reconstruct images, this method jointly recovers image latents, text embeddings, noise, and timesteps from shared gradients during fine-tuning. The proposed GradCFG framework performs a four-variable collaborative optimization, including a continuous reparameterization of the discrete timestep and a periodic noise reset to handle the non-uniqueness of noise solutions. To further enhance visual quality, Inv-Sam leverages the generative capability gap between pre- and post-fine-tuned models through a reverse-forward diffusion process to recover high-resolution details. Experiments on 512×512 DreamBooth-style fine-tuning show that the method successfully reconstructs both training images and their prompts with strong fidelity, revealing a severe privacy risk in CFG-based diffusion training.

**Strengths:**

1. This paper solves an important problem of the gradient inversion attack in classifier-free guidance diffusion models, which may actually violate the user's privacy, especially in a federated learning scenario. As a pioneering work in this field, I tend to believe it will shed light on future research.

2. The proposed function reparameterization strategy is an elegant solution, well done. By transforming the noise scheduler into a differentiable integral form, the authors eliminate the non-differentiability barrier that has prevented prior gradient-based attacks on CFG diffusion models. This design not only enables seamless optimization of $t$ alongside other latent variables, but also retains compatibility with standard diffusion scheduling. The idea is conceptually simple, mathematically well-motivated, and practically impactful—an insightful contribution that clearly advances the feasibility of gradient inversion in diffusion-based systems.

3. The four-variable joint optimization framework is well designed and technically ambitious. Prior attacks typically focus only on image reconstruction, while this work simultaneously recovers image latents, text embeddings, sampled noise, and timesteps—representing a significant expansion of attack surface and adversarial capability in multimodal diffusion training. The periodic noise reset mechanism is also a thoughtful addition, effectively addressing solution multiplicity and improving convergence stability.

**Weaknesses:**

1. **Small batch size**: The authors mainly conduct experiments with a small batch size of 5, and the images within each batch appear to focus on the same object category. While this setup is reasonable for controlled evaluation, it deviates from real-world scenarios where a batch may contain more diverse and semantically unrelated samples, making inversion substantially more challenging. Although I expect the current method may not yet fully address the inversion of highly heterogeneous batches, the proposed approaches are technically solid and represent a meaningful step forward. Despite these limitations, I still believe the contributions are significant.

2. **Lack of demonstrations of generalization**：All the experiments were conducted on TinySD. I suggest that more model types be examined, so as to show the generalizability of the proposed method.

3. **Missing Comparison with Other Gradient Inversion Methods**: Although there is no direct baseline for CFG inversion, comparing against existing inversion methods for GAN-based reconstruction attacks would strengthen the empirical claim.

**Questions:**

see weakness

---

> ### Author Response · Authors · 2025-11-22
> **Response to Reviewer TnL3 (part 1)**
>
> We are grateful for your positive assessment of the novelty and contributions of our work. We address your main concerns as follows.
>
> ### R1.1 On small batch size and concentrated sample categories
>
> Thank you for recognizing the difficulty of performing gradient inversion attacks on classifier-free guidance (CFG) models. The primary goal of our experiments is to demonstrate the **feasibility** and **potential severity** of gradient leakage on large-scale CFG models within a realistic computational budget.
>
> As you noted, prior gradient inversion works have almost exclusively focused on CNN-based classification networks or relatively small generative models [1,2,3]. In contrast, our method is built on Stable Diffusion–style CFG models [4], where:
>
> - The parameter count is larger by several orders of magnitude compared to previous works;
> - The recovery target is **multimodal** (image + text) instead of unimodal images.
>
> Under this challenging setting, the computational resource cost is particularly high, and we explicitly acknowledge this limitation in the revised manuscript.
>
> Given these constraints, our current attack scenarios do exhibit the limitations you pointed out (small batch size and concentrated categories). Nevertheless, these choices do not undermine the effectiveness of our method or its theoretical applicability to broader settings.
>
> Furthermore, the chosen scenario is not a purely toy example; it closely follows the widely used **DreamBooth-style personalization paradigm** [5], where a small number of images (around 5) from the same concept/category are used for fine-tuning. Our results show that in such realistic few-shot personalization settings, gradient information is sufficient to reconstruct both the training images and their associated text prompts, revealing a concrete privacy risk.
>
> ---
>
> ### R1.2 On generalization across model types and scales
>
> Following your suggestion, we added experiments on a **larger model**, Stable Diffusion 1.4 (SD 1.4) [4], to examine the generalization and scalability of our method. Specifically, we attack SD 1.4 to jointly recover images and text prompts, using **2 images per prompt**. The quantitative results are:
>
> *Table R1: Image reconstruction metrics on SD 1.4 for GradCFG and Inv-Sam (higher SSIM/PSNR and lower LPIPS are better).*
>
> | label       | SSIM↑ (GradCFG) | PSNR↑ (GradCFG) | LPIPS↓ (GradCFG) | SSIM↑ (Inv-Sam) | PSNR↑ (Inv-Sam) | LPIPS↓ (Inv-Sam) |
> | ----------- | --------------: | --------------: | ---------------: | --------------: | --------------: | ---------------: |
> | backpack    |          0.1568 |          10.120 |           0.7645 |          0.4511 |          12.110 |           0.5011 |
> | cat         |          0.1219 |          10.860 |           0.6761 |          0.2367 |          11.590 |           0.5165 |
> | dog         |          0.2137 |          11.180 |           0.6995 |          0.2652 |          11.250 |           0.6071 |
> | monster toy |          0.1388 |          10.550 |           0.7541 |          0.3440 |          13.320 |           0.5120 |
> | robot toy   |          0.0821 |           9.120 |           0.7400 |          0.1415 |          10.370 |           0.5623 |
> | **Average** |      **0.1427** |     **10.3660** |        **0.7268** |      **0.2877** |      **11.7280** |        **0.5398** |
>
> For text reconstruction, we report embedding similarity and the PSNR of the generated images conditioned on the recovered prompts:
>
> *Table R2: Text reconstruction metrics on SD 1.4 (higher similarity and PSNR are better).*
>
> | label       | similarity↑ |  PSNR↑ |
> | ----------- | ---------: | ----: |
> | backpack    |     0.6538 | 13.47 |
> | cat         |     0.8174 | 15.60 |
> | dog         |     0.8641 | 17.20 |
> | monster toy |     0.6311 | 12.45 |
> | robot toy   |     0.6201 | 15.34 |
> | **Average** |   **0.7173** | **14.812** |
>
> These results demonstrate that GradCFG maintains effective reconstruction quality on the larger SD 1.4 model. Although the absolute metrics are lower than on Tiny-SD, the recovered images and prompts remain semantically recognizable, which is sufficient to raise concrete privacy concerns. More detailed settings and visualizations are provided in Appendix H.

---

> ### Author Response · Authors · 2025-11-22
> **Response to Reviewer TnL3 (part 2)**
>
> ### R1.3 On baseline comparison
>
> As you observed, performing gradient inversion attacks on CFG models is substantially more difficult than on standard classifiers or small generative models. To our knowledge, there is currently **no prior work** that directly performs gradient inversion on CFG-based diffusion models with multimodal outputs, so there is no off-the-shelf baseline that exactly matches our setting.
>
> In our experiments, we identified a related method, **GIDM** [6], which performs gradient inversion on a small **unconditional** diffusion model. However, GIDM does not reconstruct text prompts, and thus cannot directly serve as a baseline for our full multimodal setting.
>
> To enable a quantitative comparison on image recovery, we adapted GIDM by providing it with the **ground-truth prompt** as prior information when reconstructing images, and we compared it to GradCFG, which **does not** access to the prompt. The results are:
>
> *Table R3: Image reconstruction comparison between the adapted GIDM baseline and GradCFG (higher SSIM/PSNR and lower LPIPS are better).*
>
> | Method                          |  SSIM↑ | PSNR↑ | LPIPS↓ |
> | ------------------------------- | -----: | ----: | -----: |
> | GIDM (with ground-truth prompt) | 0.0564 | 8.273 | 0.8566 |
> | GradCFG (without prompt)        | 0.1240 | 10.60 | 0.7778 |
>
> Even in this more challenging setting for our method, where the prompt is unknown, GradCFG still clearly outperforms the adapted GIDM baseline: SSIM improves by roughly 120%, PSNR by about 28%, and LPIPS decreases by around 10%. Full experimental details and visual comparisons are provided in Appendix I. Taken together, these results show that the effectiveness of GradCFG in this challenging CFG setting is far from trivial, as it surpasses a strong baseline that is given strictly more information.
>
>
>
> ---
>
> ### Reference
>
> [1] L. Zhu, Z. Liu, and S. Han. Deep Leakage from Gradients. NeurIPS, 2019.
>
> [2] J. Geiping, H. Bauermeister, H. Dröge, and M. Moeller. Inverting Gradients – How Easy Is It to Break Privacy in Federated Learning? NeurIPS, 2020.
>
> [3] H. Yin, A. Mallya, A. Vahdat, J. M. Alvarez, J. Kautz, and P. Molchanov. See Through Gradients: Image Batch Recovery via Gradient Inversion. CVPR, 2021.
>
> [4] R. Rombach, A. Blattmann, D. Lorenz, P. Esser, and B. Ommer. High-Resolution Image Synthesis with Latent Diffusion Models. CVPR, 2022.
>
> [5] N. Ruiz, Y. Li, V. Jampani, Y. Pritch, M. Rubinstein, and K. Aberman. DreamBooth: Fine Tuning Text-to-Image Diffusion Models for Subject-Driven Generation. CVPR, 2023.
>
> [6] J. Huang, C. Hong, S. Roos, and L. Y. Chen. GIDM: Gradient Inversion of Federated Diffusion Models. ARES, 2025.

---

> > ### Comment · Reviewer_TnL3 · 2025-11-27
> > **Response from TnL3**
> >
> > Thank you for your responses, which has adequately resolved my concerns. I’ll thereby maintain my score to support the acceptance of this paper.

---

> > > ### Author Response · Authors · 2025-11-29
> > > **Response to Reviewer TnL3**
> > >
> > > We sincerely thank you for your positive evaluation of our work. In addition, we would like to take this opportunity to further clarify and expand on one point from your initial review: you noted that our original fine-tuning experiments primarily involved images from a single object category, which could limit the perceived generality of our method. To address this concern, we conducted additional experiments to evaluate the effectiveness of our approach in a more challenging multi-category fine-tuning scenario.
> > >
> > > Specifically, we selected one image from each of several distinct object categories as fine-tuning data, and then applied both GradCFG and Inv-Sam to reconstruct the underlying images and texts. We report three standard image similarity metrics (SSIM, PSNR, LPIPS) as well as the cosine similarity between the reconstructed and ground-truth text embeddings.
> > >
> > > *Table R14: Image and text reconstruction performance of GradCFG and Inv-Sam under multi-category fine-tuning.*
> > >
> > > | Text similarity | SSIM↑ (GradCFG) | PSNR↑ (GradCFG) | LPIPS↓ (GradCFG) | SSIM↑ (Inv-Sam) | PSNR↑ (Inv-Sam) | LPIPS↓ (Inv-Sam) |
> > > | --------------- | --------------: | --------------: | ---------------: | --------------: | --------------: | ---------------: |
> > > | 0.7795          |          0.1364 |          10.460 |           0.7762 |          0.2033 |          11.520 |           0.6461 |
> > >
> > > These results show that our method remains effective even when the fine-tuning data spans different object categories: both GradCFG and Inv-Sam are able to reconstruct meaningful images and recover text embeddings with relatively high similarity (text cosine similarity ≈ 0.78). Additional visual examples and analyses for this setting are provided in Appendix M.
> > >
> > > Once again, we thank you for your recognition of our method and for highlighting this important aspect of the experimental design. We hope that these new results address your concern about diversity in the fine-tuning data, and we warmly welcome any further discussion or feedback.

---

### Author Response · Authors · 2025-11-22
**General Response**

Dear Program Chairs, Senior Area Chairs, Area Chairs, and Reviewers,

We sincerely appreciate your time, constructive feedback, and helpful suggestions. Your comments have clarified the scope of our claims and led us to strengthen the empirical validation, baseline comparisons, ablations, and presentation of GradCFG.

In response, we have carefully addressed each point and, where appropriate, added new experiments and refined existing analyses to better demonstrate the scalability and robustness of GradCFG, as well as the role of Inv-Sam. All major changes are summarized below and highlighted in the revised PDF.

---

**Additional Experiments (Weakness or Question → Revision in Revised PDF)**

1. **Baseline Comparisons**

   * As the first succesful attack on classifier-free guidance diffusion models, there is no baseline to compare. To build up a baseline, we provide the ground-truth prompts and then adapted GIDM as the baseline. The results shows that without prompot access, our GradCFG could better image recovery (TnL3, sgw7; Appendix I).

2. **Scalability to Larger Models**

   * Validation on Stable Diffusion 1.4 with joint image–prompt recovery (TnL3, sgw7, aUCK; Appendix H).

3. **Ablations**

   * Robustness over multiple random seeds (aUCK; Appendix K).
   * Ablations on the feature-mixing loss $L_{mix}$ (xW4i; Appendix J).
   * Inv-Sam variant that does not use the fine-tuned model, clarifying the additional prior it provides (xW4i; Appendix L).

---

**Clarification (Weakness or Question)**

1. Scope of claims, emphasizing DreamBooth-style personalization as the main experimental focus (xW4i).
2. Role of Inv-Sam as a refinement module that improves perceptual quality on top of gradient-based reconstructions (xW4i).
3. Further explanation of noise non-uniqueness and gradient similarity (xW4i).
4. Access to fine-tuned models in realistic collaborative/federated training scenarios (xW4i).
5. Improved organization, clearer description of challenges, and streamlined presentation throughout the main text (xW4i, aUCK).
6. Notation cleanup, clarified hyperparameters, and minor editorial fixes across the main text and appendices (all reviewers).

We welcome any further questions or suggestions from the reviewers and look forward to continued discussion.

---

### Author Response · Authors · 2025-12-02
**Summary to AC**

This note briefly summarizes the main concerns raised by the reviewers during the discussion phase and our responses, to help the AC quickly understand the key points of disagreement and our clarifications.

## 1. Baseline for the Proposed Method

Reviewers **TnL3** and **sgw7** pointed out that our experiments lack appropriate baselines for comparison.

**Our response:**

To the best of our knowledge, there is currently no existing gradient inversion attack specifically targeting CFG models, so there is no directly comparable prior baseline.

To partially address this, we adapted the **GIDM** method to our setting and, in addition, provided it with text conditions as extra prior information. Even under this favorable setup for GIDM, our method still reconstructs images with higher visual quality.

## 2. Scalability of the Method

Reviewers **TnL3**, **sgw7**, and **aUCK** raised concerns about the absence of experiments on larger-scale models; reviewers **TnL3** and **xW4i** further noted that our initial experiments focused mainly on fine-tuning with single-category images.

**Our response:**

We have added experiments on the **larger-scale** model **SD 1.4**, and the results show that our method remains effective on this model.

We have included attacks in **multi-category** fine-tuning scenarios, demonstrating that our method can still recover both image data and the corresponding text information when the personalized training data span multiple object categories.

## 3. Dependence on Access to the Fine-tuned Model

Reviewer **xW4i** questioned whether the effectiveness of the second-stage **Inv-Sam** procedure depends on having access to the attacked fine-tuned model.

**Our response:**

We introduced a variant of **Inv-Sam** that does **not** use the fine-tuned model and conducted additional experiments under this setting. The results indicate that, even without access to the fine-tuned model, this variant can still improve the semantic consistency and visual quality of the recovered data, suggesting that this stage is not strictly dependent on the fine-tuned model itself.

In addition, we note that even without the second-stage Inv-Sam procedure, the GradCFG method already achieves strong recovery performance, and Inv-Sam mainly serves to further enhance semantic alignment and perceptual quality.

## 4. Additional Ablation and Robustness Experiments

In response to requests from the reviewers for more ablation and robustness analysis, we have added and summarized the following experiments:

1. An ablation study on the feature decoupling loss $L_{\text{mix}}$; (xW4i)
2. Experiments with different random seeds and sampling timesteps for initialization; (aUCK)
3. Experiments validating the non-uniqueness of $\epsilon$ across different diffusion models. (xW4i)

---

We sincerely thank the reviewers for taking on a substantial reviewing workload and responsibility under the current circumstances. Here we have tried to summarize the main points from the discussion phase as concisely as possible, in the hope of modestly reducing your reading and decision-making burden.

---

### Meta-Review · Area_Chair_aQ6z · 2026-01-11

**Summary:**

This paper presents GradCFG, a gradient inversion attack method designed to reconstruct private training data from Classifier-Free Guidance (CFG) diffusion models like Stable Diffusion. The method jointly recovers image latents, text embeddings, noise, and timesteps from shared gradients during fine-tuning. It also introduces Inv-Sam, a refinement module that leverages differences between pre- and post-fine-tuned models to enhance semantic and perceptual quality. Experiments on DreamBooth-style personalization demonstrate the ability to jointly reconstruct images and prompts, revealing privacy risks in CFG-based training.

The reviewers raise a series of concerns covering experiment design and theoretical analysis:

1. Lack of appropriate baselines and comparisons. Reviewers TnL3 & sgw7 noted the absence of direct comparisons with existing gradient inversion methods (e.g., GIDM or GradInversion), making it hard to gauge GradCFG’s relative performance.

2. Limited scope and generalizability. Reviewers TnL3, sgw7, aUCK & xW4i questioned the experiments’ limited scale (TinySD, small batches, single-category data) and urged validation on larger models (e.g., SD 1.4), diverse categories, and varied batch compositions.

3. Dependence on fine-tuned model in Inv-Sam. Reviewer xW4i critiqued Inv-Sam’s reliance on the fine-tuned model, questioning its realism in attack scenarios and requesting clarity on whether it could work without such access.

4. Insufficient ablation and robustness analysis. Reviewers xW4i & aUCK highlighted missing ablations for hyperparameters, robustness tests across seeds/timesteps, and weak evidence for the loss’s claimed disentanglement effect.

5. Theoretical justification of Inv-Sam. Reviewer aUCK found Inv-Sam’s theoretical basis weak, noting that its improvements appear semantic rather than pixel-accurate, lacking causal linkage to model parameter shifts.

6. Overly broad claims and clarity. Reviewer xW4i felt the paper overgeneralized beyond the demonstrated DreamBooth-style personalization and noted unclear writing in parts, requiring sharper focus and scope limitation.

7. Noise non-uniqueness and optimization stability. Reviewer xW4i questioned the evidence for noise non-uniqueness, while Reviewer aUCK raised concerns about the high-dimensional optimization’s sensitivity and reproducibility.

Although the authors have addressed the concerns to some extent, the manuscript with such major concerns can hardly meet acceptance bar. Please revise the paper carefully and address the concerns.

**Reviewer Concerns:**

The rebuttal partially addressed the concerns:

1.  The rebuttal adapted GIDM with ground-truth prompts as a baseline and showed that GradCFG outperforms it even without prompt access. However, the comparison is not done on different datasets and model scales.

2. The rebuttal added experiments on Stable Diffusion 1.4, confirming GradCFG remains effective on this larger model. However, the evaluation was only conducted on the SD 1.4 and did not try other models. Why choose SD 1.4 ?

3. The rebuttal conducted new experiments with multi-class fine-tuning data, demonstrating successful reconstruction across diverse categories. However, the testing is primarily in terms of data diversity (multi-class) within a similar personalization paradigm, not fundamentally different data domains or tasks (e.g., pre-training vs. fine-tuning).

4. The rebuttal tested a variant of Inv-Sam that does not require the fine-tuned model, showing it still improves results.

5. Regarding the theoretical justification of Inv-Sam, while clarified as a “semantic refinement” module, the causal link between model differences and detail recovery remains empirical, not theoretically grounded.

6. Regarding generalization beyond personalization, despite multi-class experiments, claims about applicability to all CFG training tasks (e.g., pre-training, editing) remain extrapolative and unsupported by broader validation.

7. Regarding optimization stability & sensitivity, although seed-based tests were added, deeper analysis of convergence under varied timestep schedulers and noise configurations is still limited.

**Reviewer Scores:**

For the Reviewer TnL3: TnL3’s main concerns (baseline comparison, scalability, multi-category validation) were explicitly addressed with new experiments on SD 1.4, multi-class fine-tuning, and a GIDM baseline comparison. The reviewer explicitly stated in a follow-up comment: “Thank you for your responses, which has adequately resolved my concerns. I’ll thereby maintain my score to support the acceptance of this paper.” This indicates satisfaction and likely no downward revision.

For the Reviewer sgw7, sgw7’s primary criticisms were the lack of comparison with existing methods and validation on larger models. Although the authors provided both: a tailored comparison with GIDM (with ground-truth prompts), and new experiments on Stable Diffusion 1.4. However, these experiments in rebuttal are not comprehensive and enough for a ready paper, which may not move the reviewer toward a more positive score.

For the  Reviewer xW4i：xW4i had detailed concerns about scope, Inv-Sam dependence, noise non-uniqueness, and ablations. The authors addressed these by: （1）Clarifying scope and adding multi-category experiments. （2）Proposing an Inv-Sam variant without the fine-tuned model. （3）Expanding noise non-uniqueness analysis with SD 1.4 results. （4）Adding ablations.  However, these experiments in rebuttal are not comprehensive and enough for a ready paper, which may not move the reviewer toward a more positive score.

For the Reviewer aUCK, aUCK’s main worries were about Inv-Sam’s theoretical justification, optimization stability, and computational scalability. The authors: (1) Softened claims about Inv-Sam, reframing it as a semantic enhancer. (2) Added robustness tests across random seeds. (3) Showed SD 1.4 results to partially address scalability. However, aUCK’s final response indicated remaining concern about timestep coverage (“more timesteps settings”).

The discussion and rebuttal appear to have strengthened the paper’s standing, with at least one reviewer likely keep the acceptance threshold, improving the paper’s overall trajectory.

---

### Decision · Program_Chairs · 2026-01-26

Reject